# Rethinking Gating Mechanism in Sparse MoE: Handling Arbitrary Modality Inputs with Confidence-Guided Gate

**Liangwei Nathan Zheng** [1]  **Wei Emma Zhang** [1]  **Mingyu Guo** [1]  **Olaf Maennel** [1]  **Weitong Chen** [1]

## Abstract

Effectively managing missing modalities is a fundamental challenge in real-world multimodal learning scenarios, where data incompleteness often results from systematic collection errors or sensor failures. Sparse Mixture-of-Experts (SMoE) architecture has shown the potential to naturally handle multimodal data, with experts specializing in different modalities. However, existing SMoE approaches often lack proper ability to handle missing modality, leading to performance degradation and poor generalization in real-world applications. We propose ConfSMoE to introduce a two-stage imputation module to handle the missing modality problem for the SMoE architecture by taking the opinion of experts and revealing the insight of expert collapse from gradient analysis with strong empirical evidence. Inspired by our gradient analysis, ConfSMoE proposed a novel expert gating mechanism by detaching the softmax routing score to task confidence score w.r.t ground truth signal. This naturally relieves expert collapse without introducing additional load balance loss function. We show that the insights of expert collapse empirically align with other gating mechanism such as Gaussian and Laplacian gate. The proposed method is evaluated on four different real-world datasets with three distinct experiment settings to conduct comprehensive analysis of ConfSMoE on resistance to missing modality and the impacts of proposed gating mechanism. We have released our code in: https://github.com/IcurasLW/Official-Repository-of-ConfSMoE.git

[1]School of Computer Science and Information Technology, Adelaide University, Adelaide, Australia. Correspondence to: Weitong Chen <weitong.chen@adelaide.edu.au>.

*Proceedings of the 43rd International Conference on Machine Learning*, Seoul, South Korea. PMLR 306, 2026. Copyright 2026 by the author(s).

## 1. Introduction

As modern applications increasingly involve data from multiple heterogeneous sources, multimodal learning has emerged as a critical paradigm to unlock complementary insights and enhance decision-making in complex tasks spanning medical decision support (Zheng et al., 2024a;b; Zhai et al., 2025), cross-modal inference (Wu et al., 2017; Cheng et al., 2024), temporal reasoning (Zheng et al., 2025; Ni et al., 2024; Tan et al., 2026b), and natural language understanding (Fedus et al., 2022; Masoudnia & Ebrahimpour, 2014; Tan et al., 2026a). Sparse Mixture-of-Experts (SMoE) architectures (Yao et al., 2024; Yun et al., 2024; Mustafa et al., 2022) are widely used in multimodal learning for their ability to assign different experts to different input patterns through conditional computation. They are particularly effective when all input modalities are present, enabling diverse feature extraction and flexible expert routing. However, this assumption is often violated in practice. Incomplete modality observations are common in real-world scenarios due to sensor failures, privacy restrictions, or heterogeneous data collection pipelines (Yun et al., 2024; Sun et al., 2024; Shang et al., 2017; Wang et al., 2023). In addition, the use of softmax-based routing (Fedus et al., 2022) in typical SMoE tends to produce sharp routing distributions, leading to expert collapse problem, where only a few experts dominate the computation as shown in Figure 1a, resulting in poor diversity and limited generalization.

When one or more modalities are missing, many imputation approaches aim to reconstruct absent views through cross-modal information sharing (Ma et al., 2021; Wang et al., 2023; Yao et al., 2024) as shown in Figure 1b and overlook modality-specific information. We hypothesize that relying solely on available-modality imputation can misdirect the routing mechanism, leading to worse expert collapse; specifically, because the imputed features are dominated by the signal of available modality, the router becomes biased. For instance, if a text modality is imputed from vision, the router is likely to assign that 'pseudo-text' to a vision expert rather than a text-specific one, exacerbating expert collapse and degrading gating reliability. Therefore, current modality imputation methods degrade the reliability of the gating mechanism and result in suboptimal expert selection. Although

existing SMoE approaches attempt bypass imputation by assigning different modality combinations to designated experts (Han et al., 2024), this strategy becomes computationally intractable as the number of modalities grows. Another recent work leveraged a learnable modality bank (Yun et al., 2024) to impute missing modality by the missing pattern, and enforce load balance by load balance loss. However, the imputation bank is randomly initialized, similar to prompt tuning (Jia et al., 2022; Lester et al., 2021), and fails to capture modality-specific and fine-grained instance-specific structure, limiting the reliability of modality imputation and potentially misdirecting the router. Furthermore, enforcing load balancing via auxiliary loss functions can lead to ambiguous expert selection, as illustrated in Figure 3a.

To overcome the limited robustness of existing SMoE models under incomplete modality conditions, we propose a two-stage imputation approach based on a core insight: *a complete modality imputation should consist of modality- and instance-specific information*, a property often overlooked in prior work. Firstly, we estimate missing modalities using intra-modality correlations, empowering modality-specific information, and then refines these estimates through token-level cross-modal alignment to existing modality of the same sample, empowering instance-specific information. This refinement leverages the advantages of MoE that the data used to refine missing modality are from different opinions of experts. This approach is inspired by the modality gap phenomenon discussed by (Liang et al., 2022) and illustrated in Figure 1c, which suggests that a certain level of discrepancy between modalities is both expected and beneficial for downstream tasks. Thus, it is natural to first impute missing modalities using intra-modality information subsequently enhance them through token-level cross-modal refinement to add up instance-specific features from available modality.

We further uncover and formalize a critical cause of expert collapse in SMoE: the gradient concentration induced by softmax gating, which suppresses routing diversity and limits its generalization (Fedus et al., 2022; Wang et al., 2024a; Yun et al., 2024; Han et al., 2024; Chi et al., 2022). Through gradient analysis, we show that conventional load-balancing losses exacerbate this issue by introducing gradient conflicts during optimization, reflected by the "Sinusoidal Wave" pattern in expert selection plot shown in Figure 3a. To resolve this, we design a *confidence-guided expert routing mechanism* that replaces softmax gating with token-level confidence scores. This formulation not only avoids the adverse effects of entropy-based balancing but also enables stable, interpretable expert selection. The two mechanisms handling missing modality and Confidence-guided gating consist of our final proposed method: **ConfSMoE**. By decoupling routing score from the optimization bottlenecks of prior designs, ConfSMoE achieves both enhanced expert diversity and consistent performance across a wide range of incomplete modality scenarios. Our contributions are the following:

- A confidence-guided expert routing mechanism is proposed to mitigate expert collapse by decoupling gating from softmax-induced sharpness. Token-level confidence is used as an interpretable signal for expert selection.

- A two-stage imputation framework is developed, where missing modalities are first reconstructed using modality-specific patterns, then refined with instance-level cross-modal context to preserve structural fidelity.

- Empirical results across three missing-modality settings demonstrate strong robustness and generalization. Performance improvements are further supported by ablation studies that align with theoretical analysis.

## 2. Preliminaries

### 2.1. Mixture-of-Expert Architecture

In this work, we build upon the SMoE framework by incorporating expert routing into the Transformer's feedforward layers, enabling dynamic token-to-expert assignment based on learned routing scores. This architecture supports selective expert activation, where only the Top-K experts are engaged per token, and proves especially effective in the joint expert setting (Han et al., 2024), allowing tokens from different modalities to be routed to shared experts for enhanced cross-modal interaction. We begin with the standard Softmax SMoEs with $N$ number of expert as a baseline configuration. Denote that input token embedding $\mathbf{h} \in \mathbb{R}^d$ and the a learnable softmax router $G(\mathbf{u}) = \frac{\exp(u_i)}{\sum_j^d \exp(u_j)} = \mathbf{g} = [g_1, g_2, \ldots, g_N]$, where $\mathbf{u} = \mathbf{W}_r \mathbf{h}$ and $\mathbf{W}_r \in \mathbb{R}^{N \times d}$ is the weight of router to linearly map $\mathbf{h}$ to gating logits $\mathbf{u} \in \mathbb{R}^N$. $G(\mathbf{u})$ produces routing score $g_i$ for each expert $E_i$ from expert pool $\mathbf{E} = [E_1, E_2, \ldots, E_N] \in \mathbb{R}^{d \times d}$. We consider the Top-K operation as a binary mask $\mathbf{m} \in \{0, 1\}^N$ to select the Top-K experts in forward as shown in Eq. 1. The final tokens/samples representation is weighted summation from Top-K experts with corresponding routing score $g_i$. Formally, we can represent the forward process of SMoE as in Eq. 2:

$$\eta = sort(\mathbf{g})_K, \quad m_i = \begin{cases} 1 & \text{if } g_i \geq \eta \\ 0 & \text{otherwise} \end{cases} \tag{1}$$

$$f(\mathbf{h}) = \mathbf{h} + \sum_{i=1}^{N} m_i g_i E_i(\mathbf{h}) \tag{2}$$

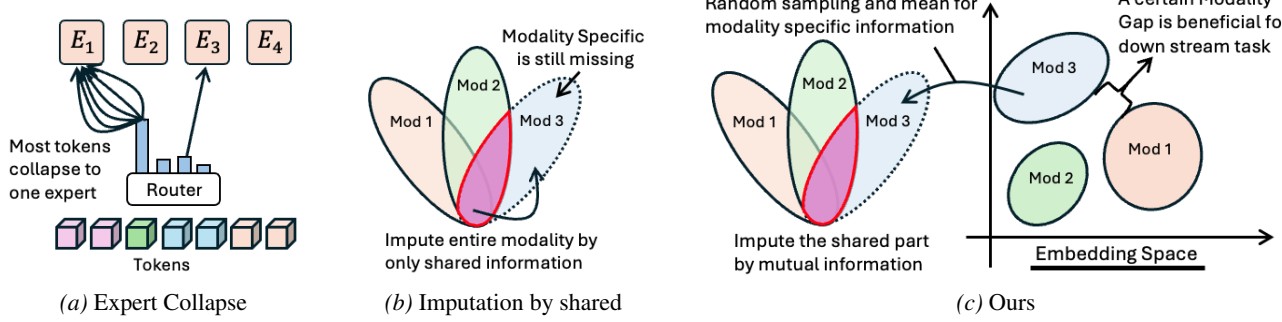

*(a)* Expert Collapse      *(b)* Imputation by shared      *(c)* Ours

*Figure 1.* Expert Collapse of MoE and Comparison with Two Different Modality Imputation

## 2.2. What Causes Expert Collapse?

Top-K operation can be considered as constant during the calculation of gradient. Thus, the Jacobian matrix of Eq. 2 w.r.t $\mathbf{h}$ can be analyzed to reveal the underlying cause of expert collapse in SMoE. Therefore,

$$\mathbf{J}_{MoE} = \sum_{i=1}^{N} m_i \frac{\partial g_i}{\partial \mathbf{h}} E_i(\mathbf{h}) + \sum_{i=1}^{N} m_i g_i E_i'(\mathbf{h}) \quad (3)$$

Note that expert $E_i(h)$ and softmax gating $G(\mathbf{u})$ themselves are differentiable w.r.t $\mathbf{h}$ and the mask $\mathbf{m} = \text{Top-K}(\mathbf{g}, K)$ is a piecewise-constant mask that only changes when the ordering of routing score $\mathbf{g}$ changes. Conditioning on a fixed mask $\mathbf{m}$, Eq. 2 reduces to a linear combination of differentiable functions (e.g. $E_i(h)$ and $G(\mathbf{u})$), where the gradient will only propagate through the terms with $m_i = 1$, enabling parameter update for those activated experts. Consequently, during training, only the selected experts are differentiable. Therefore, we can denote the Jacobian of softmax as $\mathbf{J}_{softmax} \in \mathbb{R}^{N \times N}$, where each entry $\mathbf{J}_{softmax}^{i,j} = g_i(\delta_{ij} - g_j)$ and $\delta_{ij}$ is Kronecker delta. In matrix form, Jacobian of softmax is $\mathbf{J}_{softmax} = \text{diag}(\mathbf{g}) - \mathbf{g}\mathbf{g}^\top$ and the Top-K $\mathbf{J}_{softmax} = \mathbf{m}(\text{diag}(\mathbf{g}) - \mathbf{g}\mathbf{g}^\top)$. Therefore, we can rewrite Eq. 3 to Eq. 4: In first term, only those experts with higher gating score will be chosen and the gradient is backward in $\text{diag}(\mathbf{g}) - \mathbf{g}\mathbf{g}^\top$ to learn better routing score. In the second term, the gradient is backward to those selected experts $E_i$ to learn better representation. During training, experts that receive more updates tend to be selected more frequently, as their improved representations attract higher gating scores. *This creates a feedback loop which those experts with more expressive power will lead to higher gating score and thus more updates on weight for strong expressive power, resulting expert collapse based on token preference and produce rich-get-richer expert.*

$$\mathbf{J}_{MoE} = \underbrace{\mathbf{E}(\mathbf{h})\mathbf{m}(\text{diag}(\mathbf{g}) - \mathbf{g}\mathbf{g}^\top)}_{\text{Learning better routing score}} + \underbrace{\sum_{i=1}^{N} m_i g_i E_i'(\mathbf{h})}_{\text{Learning better representation}}$$

(4)

## 2.3. Gradient of Load Balance Loss

This expert collapse phenomenon restricts the model learning capacity since many other experts are wasted and can not reflect the actual expert diversity. One widely-used solutions to achieve balance expert usage is to introduce additional load balance loss (Yun et al., 2024; Han et al., 2024). However, such load balance loss approach is difficult to optimize since it produces gradient direction opposite to routing score term in Eq 4. To show this, we consider that any load balance loss $\mathcal{L}_{load}$ will eventually result a diverse softmax routing distribution, which encourages higher entropy on the routing score distribution. From **Assumption** C.1 defined in Appendix, we define that the objective of any $\mathcal{L}_{load}$ as $\frac{1}{\mathcal{H}(\mathbf{g})}$ over the softmax distribution, where $\mathcal{H}(\mathbf{g}) = -\sum_i^N g_i \log g_i$ represent the entropy of softmax distribution $\mathbf{g}$. Taking the Jacobian over the $\mathcal{L}_{load}$ will give us (See detailed in Appendix. C.2):

$$\mathbf{J}_{load} = [\frac{1}{\mathcal{H}(\mathbf{g})^2}(log(\mathbf{g}) + 1)^\top] \cdot (\text{diag}(\mathbf{g}) - \mathbf{g}\mathbf{g}^\top) \quad (5)$$

Recall that $\mathbf{E}_K(\mathbf{h})$ in Eq.4 is a set of expert with ReLU activation, we assume this network should not output negative value as the activation is located on positive side. We also know the fact that the Jacobian of softmax $\mathbf{J}_{softmax}$ is symmetry and always Positive Semi-Definite (PSD) (Gao & Pavel, 2017), making the optimization direction of better routing score is always positive as the eigen-values are positive. In contrast, when sharp distribution appears, only one $g_j$ is closed to 1 while other $g_i, (i \neq j)$, approach to 0 in $log(\mathbf{g}) + 1 = [log(g_1) + 1, log(g_2) + 1, \dots, log(g_N) + 1]$.

Therefore, $\frac{1}{\mathcal{H}(\mathbf{g})^2}(log(\mathbf{g}) + 1)^\top$ is exponentially dominated by negative value, producing conflict gradient between $\mathbf{J}_{load}$ and the first term of $\mathbf{J}_{MoE}$. Although recent works argue that Sparse MoE is not fully differentiable (Puigcerver et al., 2023; Wang et al., 2024b) since the experts are not fully activated, the Top-K operation does not change PSD property of the first term in Eq.4. In practice, we also do not observe any optimization problem caused by this discontinuity and this observation aligns with previous research (Fedus et al., 2022). Gradient will still be backward through the term where the activation mask is $m_i = 1$ in $J_{MoE}$. In addition, we notice that first term gradient of Eq. 4 is only impacted by the Top-K $g_i$ while the gradient in Eq. 5 is dominated by the Top-K $g_i$. The sharper the gating score distribution is, the more gradient conflict will be. It is also worthy to notice that gradient in Eq.5 conflict occurs for any Top-K selection as the load balance loss consider all $g_i$ regardless of number K expert selection. Taking an extreme example, Top-1 selection (Fedus et al., 2022) generates one $g_i$ out of $\mathbf{g}$ for the second term of Eq.4 while Eq.5 still take all $g_i$ to gradient computation in practice. Therefore, *the opposite gradient from auxiliary load balance loss makes the training difficult to converge to global minimum and Expert selection becomes ambiguous, failing to reflect actual capacity of expert*.

## 3. Proposed Method

### 3.1. ConfNet:Confidence-Guided Gating Network

From Eq. 5 and Eq. 4, we understand that load balance loss produces an opposite gradient to softmax router in typical SMoE. To avoid this gradient conflict and balance the load, the potential solution is to remove the association of sharp distribution from Eq. 4 and attempt to implicitly achieve load balance. One is to replace the routing function as illustrated in Figure. 2 (c) and one is to decouple the gating score with softmax router as illustrated in Figure. 2 (d). A typical example from a recent work, FuseMoE (Han et al., 2024), they replace the Softmax router to Laplacian router and calculate the routing score based on similarity of the expert embedding by $\mathcal{L}_1$ distance. Then, the Top-K experts are selected with highest similarity to expert embedding and Laplacian router is not a sharp distribution so that the load shows implicit balance pattern as in Appendix Figure 8. The second approach is to decouple the connection of gating score and softmax distribution. One simple approach is to consider all expert equally important, then the final representation is averaged from all selected expert. However, this averaged approach does not emphasize the difference of expert and lose the specification of expert.

Therefore, we propose a Confidence-Guided Gating Network Pool (ConfNet), which introduces a novel routing strategy guided by task confidence. Specifically, instead of solely relying on conventional softmax-based gating, which often results in sharp distributions and entangled gradient contributions, ConfNet leverages the ground-truth label information to supervise the routing mechanism, encouraging expert selection aligned with task-relevant confidence rather than token or embedding similarity as in (Han et al., 2024; Fedus et al., 2022). For each expert $E_i : \mathbb{R}^d \to \mathbb{R}^d$, we introduce an auxiliary confidence estimation network: a one-layer linear network $U_i : \mathbb{R}^d \to \mathbb{R}^1$ that maps token embedding $\mathbf{h} \in \mathbb{R}^d$ to a single scalar logits $v_i \in \mathbb{R}^1$. $v_i$ is then mapped to a confidence score $c_i \in (0, 1)$ as gating score by Sigmoid, representing the confidence of ground truth label $y_t$. To learn the ground true confidence $p_t$ of downstream task w.r.t $y_t$, where $y_t$ is ground truth label. We apply MSE loss as an auxiliary loss that minimize the difference between $c_i$ and $p_t$. Formally, given any classification dataset $D$ with supervised signal $y_i$ and number of instance $|D|$, we consider the task loss $\mathcal{L}_{task}$ as Cross Entropy Loss and the final objective is to optimize:

$$\mathcal{L} = \underbrace{\frac{1}{|D|}\sum_{}^{|D|} y_t log(p_t)}_{\mathcal{L}_{task}} + \underbrace{\frac{1}{|D|K}\sum_{}^{|D|}\sum_{i=1}^{N} m_i(c_i - p_t)^2}_{\mathcal{L}_{conf}}$$

(6)

Note that while the global optimization landscape of the deep neural network remains non-convex, the selected loss components (Cross-Entropy and MSE) are smooth and convex with respect to the network outputs. This ensures that the auxiliary confidence loss provides a stable, well-behaved gradient signal during backpropagation. During both training and inference, $c_i$ serves as a proxy gating score for each expert to token fusion in Eq. 3. The $c_i$ is Token-level confidence, which each token has one corresponding $c_i$ for aggregation as in $\sum_{i=1}^{N} m_i g_i E_i(\mathbf{h})$ as in Eq. 2, resulting in a variant of proposed method for fine-grained feature learning, denoted as ConfSMoE-T. Additionally, an expert-level gating signal is also implemented for coarse-grained information retrieval. We aggregate token-level representations assigned to each expert and average them to compute expert-level gating signals as expert-level variant, denoted as ConfSMoE-E in experiment. The multimodal interaction is learned during the token assignment for all experts, which all modalities are shared one identical expert pool. This confidence-driven routing enables both fine-grained specialization and robust multimodal interaction, leading to improved generalization and interpretability.

### 3.2. Missing Modality Modeling: Two-Stage Imputation

Inspired by the modality gap phenomenon observed in contrastive learning (Liang et al., 2022), we hypothesize that a

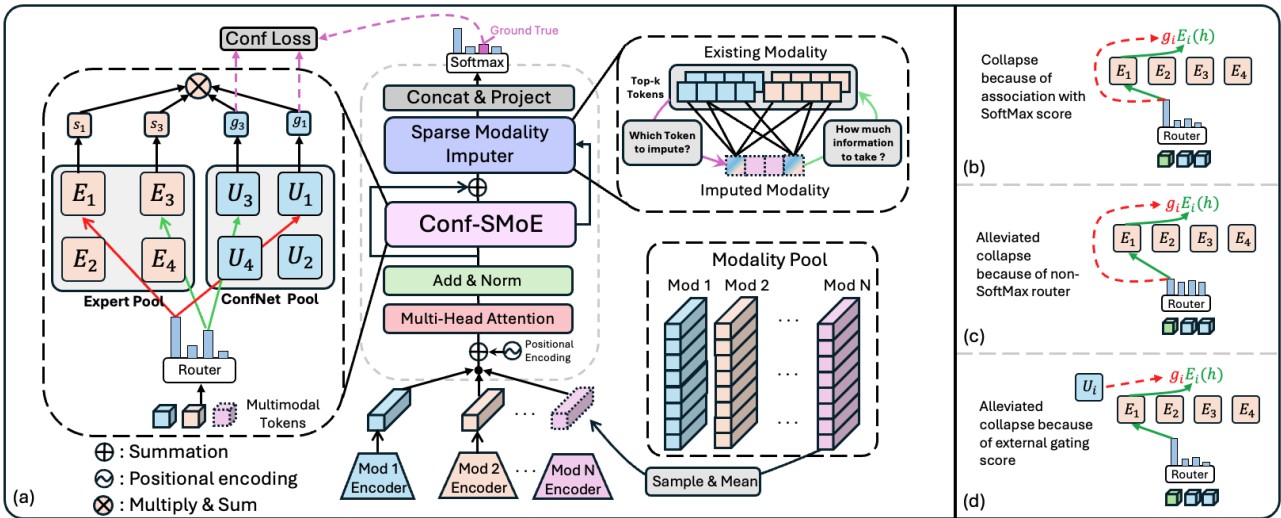

*Figure 2.* The Framework of ConfSMoE: (a) ConfSMoE considers the confidence learned from ConfNet pool as the alternative gating score for expert aggregation. The ConfNet pool is a set of one-layer linear weight $\mathbb{R}^d \to \mathbb{R}^1$. Missing modality imputation is achieved by taking the mean of random instances from the corresponding modality of training data as modality specific feature. The instance-specific feature are sparsely aggregated from existing modality. (b) Sharp gating score causes expert collapse. (c)(d) "Fair" gating score distribution and decoupling gating score can alleviate the expert collapse without the helps of load balance loss.

similar gap exists in our multimodal setting, and it may contribute positively to model performance, as illustrated in Appendix Figure 4. The presence of this modality gap suggests that imputing a missing modality using only shared information from the available modalities is inherently challenging, due to the distinct representational spaces each modality occupies. Ideally, the imputed modality should preserve the characteristics of its original distribution without being significantly biased by other modalities. To address this, we propose a **Two-stage Imputation** strategy to first reconstruct the representative feature from the missing modality distribution itself, and further refine the pre-imputed modality by adding instance-specific features from the available modalities.

**Pre-Imputation:** To retrieve modality-specific feature, given a sample $x_i$ with missing modality $M_m \in \mathbb{R}^{s \times d}$ and arbitrary available modality $M_a \in \mathbb{R}^{s \times d}$, where $s$ is the sequence length. Pre-imputation aims to obtain the modality-specific feature from other instances of $M_m$ distribution. One typical way is to take the mean feature distribution of $M_m$, but simply taking the mean may be over-fitting to training set and all representation across different modality samples will be deterministic and uninformative. In contrast to mean representation, we wish to introduce some stochasticity during the pre-imputation as this stochasticity may prevent the representation from collapsing to the same representation. Specifically, we generate an pre-imputed modality embedding $\bar{M}_m$ by sampling $n$ number of instances out of $b$ number of instances in modality pool $M_{m,i} = [M_{m,1}, M_{m,2}, \cdots, M_{m,b}]$ as follows:

$$\bar{M}_m = \frac{1}{n} \sum_{i=1}^{n} \mathbf{1} M_{m,i} \qquad (7)$$

Where $\mathbf{1} \in \{0,1\}^b$ is the selection vector for $M_m$ and $\mathbf{1}_i = 1$ represents the $i$-th instance from the modality pool that is selected. $\bar{M}_m$ will be sent to ConfSMoE to learn the interaction between different modalities. We denoted $M_m^*$ as the pre-imputed modality representation after ConfSMoE and $M_a^k$ as the available modality representation returned from k-th expert.

**Post-Imputation:** To refine $M_m^*$ by available modality, we proposed a post-modality imputation to further capture the interactions between available and missing modality. Unlike conventional approaches that solely rely on the shared representation from an individual network, our method leverages information from various experts by selectively incorporating tokens. This allows the model to absorb nuanced, instance-specific features and interaction between modalities from different views of expert to available modality. We employ a sparse attention mechanism, under the assumption that only a subset of modality-specific information is meaningfully shared across $M_a^k$. By enforcing sparsity, the model is encouraged to attend to the most relevant parts of the available modalities, thereby reducing the risk of injecting modality-irrelevant or noisy information into the imputed representation. Recall from Eq. 2, Top-K MoE generate $K$ specialized representation from available modality $M_a^k \in \mathbb{R}^{s \times d}$, $k = 0, 1, \ldots K$. Then, we concatenate all the $M_a^k$ returned from $K$ expert to form a

representation of the opinion of different experts, denoted as $M_a^* \in \mathbb{R}^{(s \times K) \times d}$. We apply a sparse cross-attention over $M_m^*$ and $M_a^*$ to select the most relevant tokens to refine $M_m^* \in \mathbb{R}^{s \times d}$. The SparseCrossAttention takes $M_m^*$ as Query modality $\mathbf{Q}_m = \mathbf{W}_q M_m^*$, $M_a^*$ as Key modality $\mathbf{K}_a = \mathbf{W}_k M_a^*$ and Value modality $\mathbf{V}_a = \mathbf{W}_v M_a^*$, where $\mathbf{Q}_m \in \mathbb{R}^{s \times d}$ and $\mathbf{K}_a, \mathbf{V}_a \in \mathbb{R}^{(s \times K) \times d}$. Note that $\mathbf{W}_q, \mathbf{W}_k, \mathbf{W}_v$ are shared between all modalities. The cross-attention scores $\mathbf{A}_a \in \mathbb{R}^{s \times (s \times K)}$ are first computed via the dot product of $\mathbf{Q}_m$ and $\mathbf{K}_a$. To induce sparsity, we apply a row-wise binary mask $\mathbf{t} \in \{0, 1\}^{s \times (s \times K)}$, where an element $t_{i,j}$ is set to 1 if the corresponding score $A_{a,i,j}$ ranks within the Top-$T$ values of the $i$-th row $\mathbf{A}_{a,i,\cdot}$, and 0 otherwise. The selection threshold is defined as $T = \frac{s(|M|-1)}{B}$, where $|M|$ represents the number of modalities in the dataset and $B$ is a sparsity hyperparameter, set to 4 by default. Finally, the sparse attention map $\mathbf{A}_a^*$ is derived through the Hadamard product of the raw scores and the binary mask as in eq.9

$$\gamma = \text{sort}(\mathbf{A}_{\mathbf{a,i},\cdot})_T, \quad t_{i,j} = \begin{cases} 1 & \text{if } A_{a,i,j} \geq \gamma \\ 0 & \text{otherwise} \end{cases} \quad (8)$$

$$\mathbf{A}_a^* = \mathbf{t} \odot softmax(\frac{\mathbf{Q_m K_a}^\top}{\sqrt{d}}) \in \mathbb{R}^{s \times (s \times K)} \quad (9)$$

Thus, we aggregate the value matrix $\mathbf{V}_a$ using the sparse attention weights $\mathbf{A}^*$ to obtain the output of $SparseCrossAttention$ (SCA). When multiple available modality $M_a^*$ are presented in the dataset, we compute their contributions individually and sum them to refine $M_m^*$:

$$M_m^* = LayerNorm \left( M_m^* + \sum^{|a|} SCA(M_m^*, M_a^*, M_a^*) \right)$$
$$= LayerNorm \left( M_m^* + \sum^{|a|} \mathbf{A}_a^* \mathbf{V}_a \right)$$

In addition, we expect this refinement to redistribute the representation of $M_m^*$ instead of generating large-scale representation and lead to divergent training. A LayerNorm layer is applied to scale the feature map back to a consistent distribution as in other layers. With SCA, the post-imputation serves as a refinement module to add an instance-specific feature to $M_m^*$, so that each modality can contribute differently on different subsegment representations of $M_m^*$. We visualize the attention weights after imputation in the Appendix Figures 14, 15, and 16, showing that the query modality can dynamically adjust the significance of information by leveraging both expert specialization and shared cross-modal cues, given that attention maps reveal distinct weight distributions across different modalities and experts.

# 4. Experiment

## 4.1. Experiment Setting

To ensure a fair comparison with baseline methods, we adopt the best-performing hyperparameter settings reported in their respective papers and official implementations. Additionally, we standardize the hidden dimension to 128 across all models. To eliminate confounding effects introduced by varying encoder architectures, we use the same modality-specific encoders for all models, as differences in encoder design can significantly influence final performance. For the MIMIC-III and MIMIC-IV datasets, we employ a pretrained ClinicalBERT (Alsentzer et al., 2019) for textual data, a one-layer patch embedding module for both irregular time series and ECG signals, and a pretrained DenseNet (Cohen et al., 2022) from TorchXRayVision for chest X-ray (CXR) images. **Experiment Setting I: Natural Missing Modality**. This setting reflects real-world missingness patterns observed in clinical datasets. **Experiment Setting II: Random Modality Dropout**. A fixed percentage of modalities are randomly dropped for each instance, while ensuring that each instance retains at least one modality in both train and test set. **Experiment Setting III: Asymmetric Modality Dropout**. Half of the modalities are randomly dropped during training, while testing is performed with only one or two modalities present per instance, simulating domain shifts or deployment constraints. All experiments are 3-Fold cross-validate in different seeds and on the same computing device for fair comparison.

## 4.2. Primary Results

**Experiment Setting I:** In Table 1 and Appendix Table 7, we provide the experiments results on experiment setting I for three different ICU benchmarking tasks, where the modalities are natural missing instead of random. We obtain the following observations from the results: (1) In clinical benchmarking tasks 48-IHM, LOS, and 25-PHE, our proposed methods can significantly outperform the most recent baseline approaches such as FuseMoE and FlexMoE and other tradidtional multimodal learning methods in terms of F1 and AUC score. For MIMIC-III, the improvement on F1 and AUC is ranging from 1.81% to 3.91% and 0.92% to 1.22% respectively. (2) The improvement is even further when the size of dataset scale to bigger dataset but the same domain like MIMIC-IV. The improvement on F1 and AUC is ranging from 1.37% to 4.12% and 1.96% to 4.85%. The above empirical evidence roots and supports our claims that the model is strongly robust to missing modality and learn more comprehensive understanding on multimodal inputs.

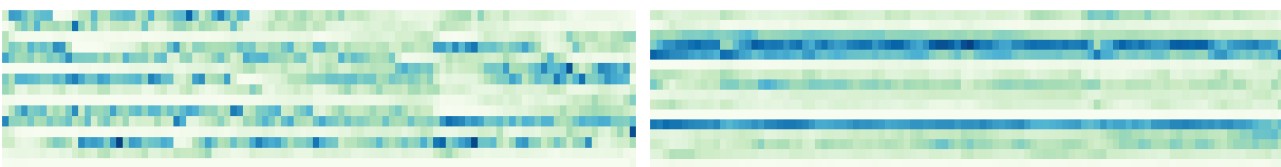

(a) Expert selection of softmax w/ $\mathcal{L}_{load}$         (b) Expert selection of ConfNet

*Figure 3.* Expert selection: X-axis represents epoch and Y-axis represents expert ID. The darker the color, the more selection will be for a particular expert. See detailed and more plots in Appendix H

*Table 1.* Experiment setting I: Main results on MIMIC-IV

| Task | Metric | SMIL | ShaSpec | mmFormer | TF | LIMoE | FuseMoE-S | FuseMoE-L | FlexMoE | **ConfSMoE-T** | **ConfSMoE-E** |
|------|--------|------|---------|----------|-----|-------|-----------|-----------|---------|----------------|----------------|
| 48-IHM | F1 | $39.58 \pm 1.12$ | $32.97 \pm 1.08$ | $45.39 \pm 1.60$ | $11.60 \pm 0.54$ | $43.76 \pm 0.44$ | $30.14 \pm 1.15$ | $40.21 \pm 1.29$ | $35.29 \pm 0.30$ | $\mathbf{49.18 \pm 0.22}$ | $48.32 \pm 0.30$ |
|  | AUC | $76.60 \pm 0.98$ | $78.29 \pm 0.25$ | $80.43 \pm 0.76$ | $67.08 \pm 0.77$ | $82.97 \pm 0.61$ | $71.34 \pm 0.88$ | $78.05 \pm 0.72$ | $80.45 \pm 0.44$ | $\mathbf{85.24 \pm 0.10}$ | $\underline{85.09 \pm 0.17}$ |
| LOS | F1 | $58.52 \pm 0.44$ | $56.22 \pm 0.28$ | $57.06 \pm 0.32$ | $42.01 \pm 1.12$ | $59.03 \pm 0.39$ | $57.59 \pm 0.76$ | $58.31 \pm 0.46$ | $56.96 \pm 0.53$ | $61.33 \pm 0.39$ | $\mathbf{61.35 \pm 0.41}$ |
|  | AUC | $75.86 \pm 0.66$ | $72.22 \pm 0.18$ | $74.65 \pm 0.32$ | $64.11 \pm 0.40$ | $76.17 \pm 0.17$ | $72.59 \pm 1.07$ | $72.59 \pm 0.61$ | $74.81 \pm 0.37$ | $\mathbf{78.22 \pm 0.15}$ | $77.85 \pm 0.12$ |
| 25-PHE | F1 | $27.77 \pm 0.91$ | $25.53 \pm 0.11$ | $26.15 \pm 0.12$ | $26.52 \pm 0.29$ | $25.38 \pm 0.52$ | $12.45 \pm 0.54$ | $12.25 \pm 0.55$ | $24.61 \pm 0.71$ | $\mathbf{28.67 \pm 0.33}$ | $28.54 \pm 0.25$ |
|  | AUC | $63.90 \pm 1.31$ | $62.57 \pm 0.23$ | $70.33 \pm 0.12$ | $56.28 \pm 0.16$ | $72.50 \pm 0.67$ | $55.48 \pm 0.39$ | $58.52 \pm 0.27$ | $71.57 \pm 0.47$ | $\mathbf{74.56 \pm 0.19}$ | $\underline{74.42 \pm 0.34}$ |

**Experiment Setting II:** In Appendix Table 8 and Table 9, we implement Random Modality Dropout to study the sensitivity of model to missing modality. In this setting, we empirically find that those methods with missing modality imputation usually shows higher tenacity in high proportion of missingness such as FlexMoE and our propose method. Although ShaSpec has also imputed the missing modality by shared information, but it is weak on the backbone model and struggles to optimize with multi-task objectives. Our proposed method, on the other hand, shows superior performance across all settings in CMU-MOSEI and CMU-MOSI for all metrics.

**Experiment Setting III:** In Appendix Table 10, Table 11, Table 12 and Table 13, we study which modality combination is significant and how is the tenacity of our proposed method to different distribution shift on missing modality. We find that single modality shows very similar performance during inference phrase in our proposed method and Flex-MoE across AUC and F1, indicating the effectiveness of missing modality imputation. In particular, our proposed also shows the highest resistance to missing modality across different combinations and remains competitive performance no matter what modality is missing. In addition, the modality combination Audio-Video is the worst modality combination for all implemented models. The potential reason may be that the data interaction between those two modalities are weak or even opposite. We show some visualization evidence in Figure 4 that the embedding of those two modalities are mostly symmetry.

### 4.3. Why Does Confidence-Guided Gating Improve Performance?

We implemented experiment on switching different gating mechanisms in our propose method as shown in Table 2 and

plot the expert selection heatmap in Appendix H. Figure 5 illustrates the softmax gating mechanism without load balance regularization, which exhibits severe expert collapse where only a few experts are consistently selected throughout training. Although load balance loss is adapted to distribute the load, it is hard to learn proper routing scores since the optimization is opposite as stated in Section 2, leading to suboptimal solution in this ablation. Softmax w/ $\mathcal{L}_{load}$ variant shows "Sharp Sinusoidal Wave" expert selection pattern as shown in Figure 3a, indicating the selection is ambiguous and difficulty on convergence. We also found that considering the uniform weight for all experts can achieve similar performance as sole softmax and generate similar sinusoidal pattern as softmax w/ $\mathcal{L}_{load}$, shown in Figure 9.

In addition, Laplacian (Han et al., 2024) and Gaussian gating (Han et al., 2024; Xu et al., 1994) function can achieve comparable to softmax w/ $\mathcal{L}_{load}$ gating mechanism without helps of $\mathcal{L}_{load}$, providing further evidence to support the advantage of balanced expert selection. Our proposed ConfNet without helps of $\mathcal{L}_{load}$ is the best and significantly outperforms other gating function in CMU-MOSI and MIMIC-III. As illustrated in Figure 6 and supported by the quantitative results in Table 2, ConfNet not only preserves expert specialization but also achieves better load balancing than Laplacian, albeit slightly less balanced than softmax with $\mathcal{L}_{load}$ and Gaussian gating as shown in Appendix H. This comparison provides two insights: First, optimization of $\mathcal{L}_{load}$ is simply against to optimization of softmax gating score but does not help to identify the best expert, reflecting the "Sharp Sinusoidal Wave" pattern in expert selection and ambiguous expert selection. Second, learning specialization of expert can also benefit to downstream performance. Notably, both ConfNet and Laplacian gates demonstrate a common pattern: *they maintain the specialization of dominant experts while redistributing some load to less active ones*, which

*Table 2.* Ablation I: Different router. CMU-MOSI adopt 50% missing setting in Experiment III

| Task | Metric | Mean | Softmax | Softmax w/ $\mathcal{L}_{load}$ | LFB | Gaussian | Laplacian | **ConfNet** |
|---|---|---|---|---|---|---|---|---|
| MIMIC-III | F1 | $48.34 \pm 0.22$ | $48.59 \pm 1.97$ | $\underline{51.67 \pm 0.62}$ | $48.36 \pm 0.88$ | $46.50 \pm 1.62$ | $50.10 \pm 1.17$ | $\mathbf{53.44 \pm 0.27}$ |
| | AUC | $85.13 \pm 0.16$ | $85.91 \pm 0.84$ | $85.97 \pm 0.52$ | $85.05 \pm 0.67$ | $\underline{86.70 \pm 0.33}$ | $85.47 \pm 1.05$ | $\mathbf{87.05 \pm 0.58}$ |
| CMU-MOSI | F1 | $42.29 \pm 0.87$ | $41.47 \pm 0.94$ | $42.43 \pm 1.34$ | $43.86 \pm 0.56$ | $42.57 \pm 1.81$ | $\underline{43.15 \pm 1.53}$ | $\mathbf{44.34 \pm 0.77}$ |
| | AUC | $68.01 \pm 1.68$ | $\underline{68.77 \pm 1.66}$ | $67.40 \pm 0.54$ | $66.60 \pm 2.24$ | $67.74 \pm 0.45$ | $67.60 \pm 1.54$ | $\mathbf{70.41 \pm 1.31}$ |

*Table 3.* Control Complexity Study: We control the complexity of model to evaluate performance

| Scale | Metrics | ShaSpec | TF | LlMoE | mmFormer | FuseMoE | FlexMoE | ConfSMoE-T |
|---|---|---|---|---|---|---|---|---|
| Small | MFLOPs | 21,73 | 4.20 | 1245.03 | 237.39 | 143.93 | 40.32 | 39,62 |
| | #Params | 2,177,406 | 2,101,571 | 2,130,227 | 2,378,371 | 20,971,020 | 2,184,963 | 2,184,730 |
| | F1 score | $39.12 \pm 1.21$ | $35.81 \pm 0.85$ | $41.11 \pm 0.90$ | $41.14 \pm 0.31$ | $36.52 \pm 0.84$ | $42.12 \pm 1.18$ | $44.63 \pm 1.65$ |
| Medium | MFLOPs | 111.27 | 45.28 | 6803,84 | 2830.37 | 286.02 | 263.94 | 268.62 |
| | #Params | 11,137,046 | 11,320,153 | 11,469,437 | 11,490,403 | 11,524,392 | 11,371,473 | 11,330,062 |
| | F1 score | $41.56 \pm 0.67$ | $38.62 \pm 0.38$ | $43.21 \pm 0.56$ | $42.44 \pm 0.79$ | $39.44 \pm 0.81$ | $43.78 \pm 0.36$ | $45.37 \pm 1.73$ |
| Large | MFLOPs | 331.87 | 133.47 | 19668.02 | 9282,03 | 308.14 | 1181.48 | 1078,82 |
| | #Params | 33,203,156 | 33,368,335 | 33,000,827 | 33,428,969 | 33,182,556 | 33,385,803 | 33,398,654 |
| | F1 score | $42.82 \pm 0.74$ | $37.64 \pm 0.88$ | $43.45 \pm 0.82$ | $41.50 \pm 0.66$ | $40.84 \pm 0.67$ | $43.46 \pm 1.36$ | $45.55 \pm 1.69$ |

highlights that strict load balancing may suppress expert specialization and degrade performance, whereas maintaining a balance between specialization and load distribution leads to superior results. We provide further explanation in Appendix H for supporting our claims.

### 4.4. How ConfNet and Modality Imputation integrate together?

We implement the experiment in Table 15 by removing ConfNet and the two-stage imputation module gradually. This experiment demonstrates that both novel modules, ConfNet and the two-stage imputation, are effective on their own, while their integration provides strong robustness against the missing modality challenge. Specifically, ConfNet captures informative features from the imputed modality, and the adaptive two-stage imputation operation reconstructs high-quality features, thereby complementing each other to achieve robust performance. Specifically, we dropped the entire imputation module and kept the missing modality as zeros (w/o impute), we can see that F1 and AUC dropped off by 3.85 and 0.74 on MIMIC-III, 1.88 and 1.97 on CMU-MOSI. This performance drop is even worse when imputation and ConfNet gating mechanism is removed from proposed method, providing strong empirical evidence for effectiveness of our design. We further remove the post-imputation (w/o post-impute) and observe that ConfNet can still take advantage of modality-specific feature for their downstream task, showing robustness against missing modality.

### 4.5. Complexity Analysis

In Table 14, we control the number of parameters in three different scales to compare performance. We aims to find the best model given similar parameter and computational costs. Although ShaSpec and TF achieve lower FLOPs given the same parameter amount, their performance remains less competitive than most Transformer- and MoE-based models. Among MoE-based models, ConfSMoE-T delivers the best performance while maintaining similar MFLOPs and parameter counts to the second-best baseline. Under comparable parameter settings, LIMoE and mmFormer fail to keep FLOPs as low as ConfSMoE-T and yield suboptimal results. Moreover, LIMoE is particularly inefficient, as its MFLOPs are substantially higher than those of other models with similar parameter sizes. By increasing the number of parameter to a larger scale, we can observed different amount of increment across the selected models while none of the baselines can achieve the same performance as ConfSMoE-T. Many baselines are still less competitive to small-scale ConfSMoE-T, demonstrating the scalability of ConfSMoE.

## 5. Related Works

While many works consider that the interaction between multimodal data can bring a comprehensive understanding for downstream task. Initial multimodal fusion approach usually incorporates neural kernel fusion(Bucak et al., 2013; Poria et al., 2015), early/late network fusion (Xu et al., 2023; Baltrušaitis et al., 2018; Guo et al., 2019). Approaches such as Tensor Fusion Network(Zadeh et al., 2017), Multimodal Transformer(Tsai et al., 2019) and Multimodal Adaption Gate(Rahman et al., 2020), LlMoE(Mustafa et al., 2022)

all highlights the multimodal interactions are beneficial to downstream tasks. In medical domain, MedFuse (Hayat et al., 2022) incorporated CXR and EHR modality to jointly fuse modality with early and later fusion in a single fusion network. MISTS (Zhang et al., 2023) incorporated multitime attention mechanism to encode irregular time serires and adapts multimodal cross attention to fuse two different modality. (Yang & Wu, 2021) fused multimodal data in a attention gate and compute a replacement vector for modality fusion. However, these approaches are limited to full modality only and difficult to extend to increasing number of modality. Missing modality is realistic and practical problem in real life application such as medical diagnosis. SMIL (Ma et al., 2021) proposed a Bayesian meta-learning solution with reconstruction network to impute the missing modality from in hidden state during fusion. (Du et al., 2018; Shang et al., 2017) propose generative adversarial networks to impute the missing modality by semi-supervised learning. ShaSpec (Wang et al., 2023) and DrFuse (Yao et al., 2024) designed multiple loss functions to learn share and specific representation of different modality. However, the reconstruction network from SMIL and ShaSpec requires multiple additional auxiliary loss functions, which occupies the great amount of gradient in optimization process instead of task specific loss. (Sun et al., 2024) In addition, recent studies like FuseMoE (Han et al., 2024) respect the missingness of modalities and wish to assign each combination of multimodal data to different experts. Flex-MoE(Yun et al., 2024) further expand the advantages of FuseMoE and introduce a modality bank to impute the missing modality.

## 6. Conclusion

In this work, a novel ConfSMoE model, a confidence-guided sparse mixture-of-experts framework, is proposed to robustly handle multimodal learning with missing modalities. From gradient analysis, we show that the traditional load-balanced loss can cause gradient conflict with learning a better routing behaviour in the Sparse MoE architecture. Therefore, we proposed two key innovations: a two-stage imputation strategy that preserves modality-specific structure, and a novel confidence-driven gating mechanism that decouples expert selection from softmax-induced sharpness. These components jointly improve both expert specialisation and routing stability without requiring entropy-based balancing losses.

## Impact Statement

The goal of this work is to advance the field of Machine Learning and its applications on imperfect data. Because our methodology relies exclusively on privacy-neutral datasets, we deliver a high-impact technical contribution that is inherently free of societal or ethical risk.

## Acknowledgment

This work was supported by Australian Research Council ARC Early Career Industry Fellowship (Grant No.IE240100275), Australian Research Council Discovery Projects (Grant No.DP240103070), Australian Research Council Linkage(Grant No.LP230200821), 2026 Global Partnership Joint Fund and Australia's Economic Accelerator Ignite (Grant No. IG250200014).

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

## A. Impact Statements

This paper presents work whose goal is to advance the field of machine learning. There are many potential societal consequences of our work, none of which we feel must be specifically highlighted here

## B. Notation

*Table 4.* Notation

| Variable | Definition |
|---|---|
| $E_i$ | i-th expert function |
| $\mathbf{E}$ | Expert pool |
| $\mathbf{U}$ | ConfNet pool |
| $\mathbf{h}$ | Embedding of arbitrary data input |
| $K$ | Number of K selection |
| $T$ | Number of T selection in Post-Imputation |
| $N$ | Total number of expert |
| $n$ | Number of instances selected in Pre-Imputation |
| $G(\cdot)$ | Gating function |
| $\mathbf{g}$ | Gating score vector |
| $\mathcal{H}(\cdot)$ | Information entropy |
| $p_t$ | Confidence of ground truth |
| $\mathbf{A}_a^*$ | Sparse attention map |
| $M_m$ | Representation of Missing modality $m$ before model backbone |
| $M_{m,b}$ | Modality pool with $b$ instance for $M_m$ |
| $\bar{M}_m$ | Imputed representation of $M_m$ after pre-imputation |
| $M_m^*$ | Imputed representation of $M_m$ after expert forward and Post-imputation |
| $M_a$ | Representation of available modality $a$ before model backbone |
| $M_a^k$ | Representation of $M_a$ returned from k-th expert |
| $M_a^*$ | Representation of $M_a$ after expert forward |
| $B$ | Sparsity hyperparameter |
| $b$ | Number of instance in modality pool |
| $\|M\|$ | Number of modality |
| $\mathbf{diag}(\cdot)$ | Diagonal matrix |
| $\delta_{ij}$ | Kronecker delta |
| $u_i$ | Softmax logits input at i-th entry |
| $c_i$ | Confidence score for expert $E_i$ |
| $g_i$ | Gating score for expert $E_i$ |

To facilitate understanding of the model formulation and derivations, Table 4 outlines the notations used throughout this paper. These include variables associated with expert routing, gating mechanisms, modality assignments, and sparsity control, which are central to our proposed ConfSMoE framework.

## C. Derivation of Jacobian for MoE and Load Balance Loss

### C.1. Jacobian for Softmax

This section provides detailed mathematical derivations of the Jacobian matrix used in the gating function of ConfSMoE and its relation to the load balance loss. We first derive the Jacobian of the softmax operation applied to the gating logits, which is essential for analyzing expert selection dynamics. We then extend this analysis to derive gradients of the entropy-based load balance loss. These derivations provide theoretical insights into how sparse expert routing and balanced expert utilization can be jointly optimized.

We first derive the Jacobian of the softmax function as used in the MoE formulation Eq. 2.

$$\mathbf{g} = G(\mathbf{u}) = softmax(\mathbf{u}) = \frac{e^{u_i}}{\sum_{j=1}^{N} e^{u_j}} \tag{10}$$

where $u_i$ is the i-th input value of gate logits.

$$\frac{\partial \mathbf{g}}{\partial \mathbf{u}} = \begin{cases} \dfrac{e^{u_i}(\sum_{j=1}^{N} e^{u_j} - e^{u_i})}{(\sum_{j=1}^{N} e^{u_j})^2} & \text{if } i = j \\[3mm] -\dfrac{e^{u_i} e^{u_j}}{(\sum_{j=1}^{N} e^{u_j})^2} & \text{if } i \neq j \end{cases} \tag{11}$$

$$\frac{\partial \mathbf{g}}{\partial \mathbf{u}} = \begin{cases} g_i(1 - g_j) & \text{if } i = j \\ -g_i g_j & \text{if } i \neq j \end{cases} \tag{12}$$

Where $g_i$ represent $\frac{e^{u_i}}{\sum_{j=1}^{N} e^{u_j}}$. This can be further compactly represent as:

$$\frac{\partial \mathbf{g}}{\partial \mathbf{u}} = g_i(\delta_{ij} - g_j) \tag{13}$$

Where $\delta_{ij} = \begin{cases} 1 & \text{if } i = j \\ 0 & \text{if } i \neq j \end{cases}$ is Kronecker delta and the matrix form is:

$$\mathbf{J}_{softmax} = \frac{\partial \mathbf{g}}{\partial \mathbf{u}} = \mathbf{diag}(\mathbf{g}) - \mathbf{g}\mathbf{g}^\top \tag{14}$$

### C.2. Jacobian for Load Balance Loss

**Assumption C.1.** For any load balance loss $\mathcal{L}_{\text{load}}$ applied to a Softmax-based MoE, the effect of $\mathcal{L}_{\text{load}}$ is to encourage a flat expert selection distribution, which is equivalent to maximizing the entropy of the gating scores.

With **Assumption** C.1, We have load balance loss $\mathcal{L}_{load} = \frac{1}{\mathcal{H}(\mathbf{g})}$, where $\mathcal{H}(\mathbf{g}) = -\sum_{i=1}^{N} g_i log(g_i)$ is the entropy of the gating distribution.

By applying the chain rule, we obtain:

$$\frac{\partial \mathcal{L}_{load}}{\partial \mathbf{u}} = -\frac{1}{\mathcal{H}(\mathbf{u})^2} \frac{\partial \mathcal{H}}{\partial \mathbf{u}} = -\frac{1}{\mathcal{H}(\mathbf{u})^2} \frac{\partial \mathcal{H}}{\partial \mathbf{g}} \frac{\partial \mathbf{g}}{\partial \mathbf{u}} \tag{15}$$

The intermediate gradients are given by:

$$\frac{\partial \mathcal{H}}{\partial \mathbf{g}} = -log(\mathbf{g}) - \mathbf{g}\frac{1}{\mathbf{g}} = -log(\mathbf{g}) - 1, \quad \frac{\partial \mathbf{g}}{\partial \mathbf{u}} = \mathbf{diag}(\mathbf{g}) - \mathbf{g}\mathbf{g}^\top \tag{16}$$

Therefore, he final Jacobian of the load balance loss $J_{load}$ is:

$$\mathbf{J}_{load} = \frac{\partial \mathcal{L}_{load}}{\partial \mathbf{g}} = \left[ \frac{1}{\mathcal{H}(\mathbf{g})^2} (log\mathbf{g} + 1)^\top \right] \cdot (\mathbf{diag}(\mathbf{g}) - \mathbf{g}\mathbf{g}^\top) \tag{17}$$

## D. Experiment Details

### D.1. Dataset

**MIMIC-III** (Johnson et al., 2016) is a public dataset contains irregular time series and clinical notes modality. We follow the preprocessing scripts from MMIMI-III benchmark (Harutyunyan et al., 2019) and (Zhang et al., 2023) to form multimodal

patient instance. However, (Zhang et al., 2023) filter the instances with missing modality and we modified the scripts to preserve with instances with missing modality. **MIMIC-IV** (Johnson et al., 2023) is a public dataset that contains irregular time series, clinical notes, ECG signal and chest image. We follow the preprocessing scripts from MedFuse (Hayat et al., 2022) to preprocess irregular time series and further preprocess clinical notes, ECG signal and chest image by ourself. We follow the dataset split in MedFuse (Hayat et al., 2022) and form multimodal patient instances. **CMU-MOSI** (Zadeh et al., 2016) is a multimodal sentiment analysis dataset comprising audio, video, and text modalities. **CMU-MOSEI** (Bagher Zadeh et al., 2018) is an extension of the CMU-MOSI dataset, utilizing the same modalities—audio, text, and video from YouTube recordings. We follow preprocessing script of CMU-MOSI and CMU-MOSEI from MMML(Mustafa et al., 2022).

### D.2. Baselines

**SMIL**(Ma et al., 2021) impute missing modality with Bayesian meta-learning. **ShaSpec**(Wang et al., 2023) leverages multiple loss function to learn the modality-share and modality-specific information and impute the missing modality with modality-shared information. **TF**(Zadeh et al., 2017) extract the modality interaction by multimodal sub-networks and tensor fusion layer. **mmFormer**(Tsai et al., 2019) is a typical multimodal transformer with multihead attention mechanism. **LlMoE**(Mustafa et al., 2022) achieves the multimodal learning with contrastive learning. **FuseMoE**(Han et al., 2024) respect the missing modality and jointly learn single and missing modality combination data input and FuseMoE implement Laplacian gating mechanism for modality fusoin and expert selection. In experiment, we denote FuseMoE-S as softmax gating and FuseMoE-L as Laplacian gating. **FlexMoE**(Yun et al., 2024) assigns the combination of modality to different experts and impute the missing modality by modality bank. For our approach, we use ConfSMoE-T denotes Token-Level confidence fusion and ConfSMoE-E denotes Expert-level confidence fusion.

### D.3. Dataset and Task Description

*Table 5.* Dataset statistics

| Dataset | Task | #Samples | Modality | #Classes | Is Imbalanced? | Missing Ratio |
|---|---|---|---|---|---|---|
| MIMIC-III | 48-IHM | 6621 | Time Series, Text | 2 | Yes | 24.21% |
| | LOS | 6621 | Time Series, Text | 2 | Yes | 24.21% |
| | 25-PHE | 12,278 | Time Series, Text | 25 | Yes | 24.25% |
| MIMIC-IV | 48-IHM | 22,733 | Image, ECG, Time Series, Text | 2 | Yes | 47.10% |
| | LOS | 22,733 | Image, ECG, Time Series, Text | 2 | Yes | 47.10% |
| | 25-PHE | 47,027 | Image, ECG, Time Series, Text | 25 | Yes | 50.32% |
| CMU-MOSI | Sentiment Analysis | 1870 | Audio, Video, Text | 3 | Yes | 0% - 50% |
| CMU-MOSEI | Sentiment Analysis | 20,985 | Audio, Video, Text | 3 | Yes | 0% - 50% |

**48-IHM** is a binary classification task, where the time series is truncated to 48 hours readings and predict whether the patient in-hospital mortality in ICU.

**LOS** is also binary classification task. We formulate task similar to 48-IHM and those patients who spent at least 48 hours in the ICU are filtered to predict the remaining Length Of Stay.

**25-PHE** is a multilabel classification problem. We attempt to predict one 25 acute care conditions presented in a givem ICU stay record. The definition of those 25 acute care conditions are defined in MIMIC benchmark (Johnson et al., 2016)

**CMU-MOSI** and **CMU-MOSEI** are multimodal sentiment analysis benchmarks consisting of video clips annotated with sentiment scores. Each sample includes aligned audio, video, and text modalities. To simulate modality missingness, we introduce controlled random dropout during training.

All datasets are publicly available and widely adopted in multimodal research. Those datasets can be found and downloaded from the following links:

- CMU-MOSI and CMU-MOSEI download link: https://github.com/zehuiwu/MMML

- MIMIC-III download link: https://physionet.org/content/mimiciii/1.4/ and benchmark: https://github.com/YerevaNN/mimic3-benchmarks

- MIMIC-IV download link: https://physionet.org/content/mimiciv/2.2/ and benchmark: https://github.com/nyuad-cai/MedFuse

## D.4. Hyperparameter and Hardware Setting

All experiments are conducted in a remote server with $4 \times$ RTX4090 24 GB with 256 GB RAM and 96-cores CPU.

*Table 6.* Hyperparameter setting

| Parameter | Value |
|---|---|
| #Expert | 8 for MIMIC-III and MIMIC-IV, 4 for CMU-MOSI and CMU-MOSEI |
| #Top-K Expert | 2 |
| Learning Rate | 3e-4 |
| Hidden Size | 128 |
| #MoE layers | 1 |
| Training Epoch | 50 |
| Weight for $\mathcal{L}_{conf}$ | 1 |
| #Instance for Pre-imputation | 10 |
| Random Seed | 2023, 2024, 2025 |
| Dropout | 0.1 |

## E. Other Gating Mechanism and Load Balance Loss

### E.1. Mean Gating

The Mean Gate considers the gating score for all experts equally important. Given gating weight $\mathbf{W}_r$, and arbitrary representation $\mathbf{h}$ and gating score $g_i = \frac{1}{N}$ produced by router $G(\mathbf{u})$, where $N$ is the number of expert.

### E.2. Softmax Gating

Given a token embedding $\mathbf{h} \in \mathbb{R}^d$ and there are $N$ expert $E_1, E_2, \cdots, E_N$ with gating scores collected in $\mathbf{g} = [g_1, g_2, \cdots, g_N]$. We introduce a linear weight $\mathbf{W}_r$ to map $\mathbf{h}$ to expert logits $\mathbf{u} = \mathbf{W}_r \mathbf{h} \in \mathbb{R}^N$. As in expert selection, we use softmax to convert the logits to probability distribution:

$$g_i = \frac{\exp(u_i)}{\sum_{j=1}^{N} \exp(u_j)}.$$

### E.3. Loss-Free Balance Gating

As the Loss-Free Balance Gating (LFB) (Wang et al., 2024a) is not officially open-source, we found a github repository that reproduces the implementation: https://github.com/ambisinister/lossfreebalance. Formally, the number of instances assigned to each expert $z_i = [z_1, z_2, \cdots, z_i]$ and the average number of instance $\bar{z}$. We introduce a expert-wise bias term $\{b_i\}_{i=1}^{N}$ initialized as zeros and update iteratively in a rule-based manner. This bias term will be added to gating logits $\mathbf{u} = \mathbf{W}_r \mathbf{h} \in \mathbb{R}^N$ of each expert to get a biased gating score $u_i^*$.

$$u_i^* = u_i + b_i, \quad b_i = b_i + \alpha \times sign(e_i), \quad e_i = \bar{z} - z_i \tag{18}$$

Where, $e_i$ is a load violation error and $\alpha$ is hyperparameter to control the strength of adjustment from violation error. Note that this $u_i^*$ is only used for expert selection in Top-K function instead of expert forward. The gating score $g_i$ used to forward is generated as $g_i = Sigmoid(u_i)$. The authors used sigmoid as they believe the sigmoid achieve slightly better performance than softmax.

### E.4. Gaussian Gating

We follow the implementation of FuseMoE (Han et al., 2024) for the Gaussian gate baseline. Given a token embedding $\mathbf{h} \in \mathbb{R}^d$ and there are $N$ expert $E_1, E_2, \cdots, E_N$ with gating scores collected in $\mathbf{g} = [g_1, g_2, \cdots, g_N] \in \mathbb{R}^N$. We introduce a matrix of Gaussian gate centers, a learnable parameter denoted as $\mathbf{c} = [c_1, c_2, \cdots, c_N] \in \mathbb{R}^{d \times N}$ for each expert. For a single token, we define the gaussian logits as negative square Euclidean distance $\mathbf{h}$ and each gate center $c_i$:

$$u_i = -\|\mathbf{h} - c_i\|_2^2, \quad i = 1, \cdots, N.$$

As in expert selection we still use softmax to convert the logits to probability distribution:

$$g_i = \frac{\exp\left(-\|\mathbf{h} - c_i\|_2^2\right)}{\sum_{j=1}^N \exp(-\|\mathbf{h} - c_j\|_2^2)}.$$

### E.5. Laplacian Gating

We follow the implementation of FuseMoE (Han et al., 2024) for the Laplacian gate baseline. Given a token embedding $\mathbf{h} \in \mathbb{R}^d$ and $N$ experts $E_1, E_2, \cdots, E_N$ with gating scores collected in $\mathbf{g} = [g_1, g_2, \cdots, g_N] \in \mathbb{R}^N$, we introduce a matrix of Laplacian gate centers, a learnable parameter denoted as $\mathbf{c} = [\mathbf{c}_1, \mathbf{c}_2, \cdots, \mathbf{c}_N] \in \mathbb{R}^{d \times N}$ for each expert. For a single token, we define the Laplacian logits as the negative Euclidean distance between $\mathbf{h}$ and each gate center $\mathbf{c}_i$:

$$u_i = -\|\mathbf{h} - \mathbf{c}_i\|_2, \quad i = 1, \cdots, N.$$

As in expert selection, we still use softmax to convert the logits into a probability distribution:

$$g_i s = \frac{\exp\left(-\|\mathbf{h} - c_i\|_2\right)}{\sum_{j=1}^N \exp\left(-\|\mathbf{h} - c_j\|_2\right)}.$$

### E.6. Load Balance Loss

We follow (Shazeer et al., 2017) to implement load balance loss used in our experiment.

$$\mathcal{L}_{balance} = \mathbf{CV}^2 \left( \sum_j^N importance_j \right) + \mathbf{CV}^2 \left( \sum_j^N load_j \right)$$

$$importance_j = \sum_i^{|D|} g_{i,j}, \quad load_j = \sum_i^{|D|} \delta(g_{i,j} > 0)$$

where $CV^2(x) = \left( \frac{\sigma(x)}{\mu(x)} \right)^2$, $\sigma(x)$ is the standard deviation of $x$, $importance_j$ is the expert importance of expert $j$ and $load_j$ is load of expert $j$, $\delta(\cdot > 0)$ is an indicator function that is 1 when the inner value is greater than 0.

Now we show that the load balance function used in the experiment encourages flat gating score distribution, which is equivalent to having the same mathematical behaviour as the general load balance loss in gradient analysis. The denominator is the average across all expert, then we have:

$$\mu = \frac{1}{N} \sum_j^N importance_j = \frac{1}{N} \sum_j^N \left( \sum_i^{|D|} g_{i,j} \right) = \frac{1}{N} \sum_i^{|D|} \left( \sum_j^N g_{i,j} \right) = \frac{|D|}{N} \tag{19}$$

Note that $\sum_j^N g_{i,j} = 1$ is the summation of gating score out of softmax router. From above, the denominator is a constant only impacted by the number of instances $|D|$ and the number of experts $N$. Thus, the applied load balance loss will minimize the variance of importance for each expert, which encourages flat gating score distribution. This is equivalent to minimizing the reverse of gating score entropy $\frac{1}{\mathcal{H}(\mathbf{g})}$ as we show in gradient analysis.

# F. Limitation

This work may be limited to expert discontinuity as it is build upon on sparse MoE backbone. A key limitation lies in the dichotomy between fully differentiable MoE and sparse MoE architectures. These two paradigms embody opposite design philosophies, one prioritizing differentiability and comprehensive training, the other emphasizing efficiency through sparsity, making it difficult to simultaneously leverage the benefits of both within a single unified framework.

# G. Additional Results

These experiments extend the main results presented in Section Experiment and highlight the scalability of our method to real-world multimodal challenges.

*Table 7.* Experiment setting I: Main results on MIMIC-III

| Task | Metric | SMIL | ShaSpec | mmFormer | TF | LlMoE | FuseMoE-S | FuseMoE-L | FlexMoE | ConfSMoE-T | ConfSMoE-E |
|---|---|---|---|---|---|---|---|---|---|---|---|
| 48-IHM | F1 | 48.56 ± 0.35 | 34.36 ± 0.73 | 43.80 ± 0.67 | 23.09 ± 1.13 | 47.62 ± 0.21 | 48.62 ± 0.27 | 51.63 ± 0.39 | 47.81 ± 0.31 | **53.44 ± 0.27** | 50.00 ± 2.17 |
| | AUC | 84.00 ± 1.10 | 79.86 ± 0.53 | 81.27 ± 0.76 | 78.47 ± 0.65 | 83.30 ± 0.64 | 85.81 ± 0.13 | 86.22 ± 0.15 | 85.10 ± 0.61 | 87.05 ± 0.58 | **87.14 ± 0.21** |
| LOS | F1 | 62.01 ± 0.54 | 57.27 ± 0.36 | 61.10 ± 0.65 | 56.23 ± 0.79 | 62.75 ± 0.10 | 62.26 ± 0.36 | 62.27 ± 1.61 | 64.18 ± 0.34 | 66.27 ± 0.40 | **66.39 ± 0.23** |
| | AUC | 80.08 ± 0.35 | 74.71 ± 0.07 | 77.94 ± 0.12 | 72.60 ± 0.79 | 79.80 ± 0.27 | 80.05 ± 0.51 | 79.70 ± 0.43 | 80.49 ± 0.48 | 81.66 ± 0.26 | **81.71 ± 0.18** |
| 25-PHE | F1 | 33.01 ± 0.30 | 19.52 ± 0.39 | 35.10 ± 0.38 | 30.38 ± 0.38 | 36.22 ± 0.39 | 33.48 ± 0.22 | 34.07 ± 0.51 | 35.31 ± 0.57 | 37.40 ± 0.13 | **40.13 ± 0.40** |
| | AUC | 75.17 ± 1.84 | 64.61 ± 0.25 | 71.79 ± 0.26 | 66.56 ± 0.17 | 77.93 ± 0.44 | 77.22 ± 0.29 | 77.19 ± 0.33 | 77.37 ± 0.10 | **78.93 ± 0.38** | 78.28 ± 0.73 |

*Table 8.* Experiment setting II: Main results on CMU-MOSI

| Missing Rate | Metric | ShaSpec | TF | mmFormer | LlMoE | FuseMoE-S | FuseMoE-L | FlexMoE | ConfSMoE-T | ConfSMoE-E |
|---|---|---|---|---|---|---|---|---|---|---|
| 0% | F1 | 41.78 ± 0.64 | 43.28 ± 0.36 | 50.22 ± 0.20 | 50.91 ± 0.19 | 47.69 ± 0.39 | 48.15 ± 0.57 | 51.50 ± 0.55 | **51.83 ± 0.27** | 51.70 ± 0.29 |
| | AUC | 65.08 ± 0.39 | 64.48 ± 0.51 | 74.62 ± 0.49 | 78.12 ± 0.31 | 75.26 ± 0.20 | 75.96 ± 0.29 | 79.38 ± 0.23 | 80.56 ± 0.10 | **80.62 ± 0.07** |
| 10% | F1 | 43.40 ± 0.74 | 40.14 ± 0.13 | 48.66 ± 1.20 | 49.91 ± 0.33 | 46.81 ± 0.45 | 48.81 ± 0.42 | 49.92 ± 0.28 | 49.99 ± 0.85 | **50.89 ± 0.07** |
| | AUC | 65.45 ± 0.56 | 61.54 ± 0.66 | 77.21 ± 0.76 | 77.12 ± 0.14 | 70.25 ± 0.93 | 72.36 ± 0.50 | 77.07 ± 0.21 | **78.06 ± 0.64** | 78.03 ± 0.21 |
| 20% | F1 | 39.15 ± 0.13 | 42.64 ± 0.30 | 48.51 ± 1.15 | 48.84 ± 0.61 | 47.92 ± 0.58 | 45.00 ± 0.44 | 49.16 ± 0.24 | **50.58 ± 0.73** | 50.17 ± 0.31 |
| | AUC | 60.87 ± 0.29 | 65.52 ± 0.98 | 74.16 ± 0.71 | 76.64 ± 0.23 | 71.05 ± 0.36 | 72.15 ± 0.37 | 73.80 ± 1.43 | 77.17 ± 0.55 | **77.18 ± 0.11** |
| 30% | F1 | 39.54 ± 0.17 | 40.42 ± 0.75 | 46.66 ± 1.24 | 46.13 ± 0.55 | 43.78 ± 0.51 | 44.48 ± 0.63 | 45.49 ± 0.44 | **47.57 ± 0.10** | 47.52 ± 0.77 |
| | AUC | 60.80 ± 0.44 | 60.57 ± 0.99 | 71.25 ± 0.57 | 74.33 ± 0.49 | 70.38 ± 0.56 | 72.25 ± 0.48 | 72.54 ± 0.92 | **75.59 ± 0.18** | 75.42 ± 0.49 |
| 40% | F1 | 35.32 ± 0.32 | 38.69 ± 0.42 | 42.22 ± 0.44 | 43.31 ± 0.38 | 41.88 ± 0.42 | 43.20 ± 0.20 | 43.13 ± 0.61 | **45.12 ± 0.27** | 44.18 ± 0.10 |
| | AUC | 59.58 ± 0.13 | 61.62 ± 0.07 | 68.44 ± 0.14 | 67.21 ± 0.43 | 64.61 ± 0.63 | 66.14 ± 0.11 | 68.12 ± 0.34 | **71.65 ± 0.47** | 69.55 ± 0.49 |
| 50% | F1 | 36.39 ± 0.97 | 38.46 ± 1.09 | 41.14 ± 0.31 | 41.54 ± 0.77 | 38.55 ± 0.34 | 40.90 ± 0.49 | 41.95 ± 0.48 | **44.34 ± 0.77** | 42.48 ± 0.12 |
| | AUC | 55.60 ± 0.12 | 60.49 ± 0.80 | 61.87 ± 0.67 | 67.43 ± 0.34 | 62.28 ± 0.28 | 65.73 ± 0.30 | 66.47 ± 0.49 | **70.41 ± 1.31** | 69.04 ± 0.32 |

*Table 9.* Experiment setting II: Main results on CMU-MOSEI

| Missing Rate | Metric | ShaSpec | TF | mmFormer | LlMoE | FuseMoE-S | FuseMoE-L | FlexMoE | ConfSMoE-T | ConfSMoE-E |
|---|---|---|---|---|---|---|---|---|---|---|
| 0% | F1 | 47.48 ± 0.27 | 38.89 ± 0.54 | 56.75 ± 0.47 | 59.13 ± 0.52 | 57.04 ± 0.15 | 58.71 ± 0.23 | 60.62 ± 0.96 | 61.31 ± 0.14 | **62.35 ± 0.12** |
| | AUC | 67.93 ± 0.56 | 64.26 ± 0.61 | 76.42 ± 0.24 | 80.29 ± 0.31 | 78.54 ± 0.42 | 78.13 ± 0.34 | 81.34 ± 0.20 | **82.87 ± 0.10** | 82.75 ± 0.44 |
| 10% | F1 | 47.41 ± 0.18 | 34.94 ± 1.10 | 55.37 ± 0.47 | 57.46 ± 1.10 | 55.90 ± 0.55 | 56.68 ± 0.31 | 58.66 ± 1.10 | 59.81 ± 0.34 | **60.22 ± 0.31** |
| | AUC | 69.26 ± 0.22 | 61.05 ± 0.34 | 75.36 ± 0.53 | 79.38 ± 0.94 | 76.38 ± 0.39 | 76.44 ± 0.12 | 79.48 ± 0.23 | 81.00 ± 0.10 | **81.05 ± 0.17** |
| 20% | F1 | 46.54 ± 1.04 | 35.37 ± 0.94 | 52.91 ± 0.75 | 55.35 ± 0.88 | 54.44 ± 0.27 | 54.60 ± 0.19 | 55.96 ± 1.67 | **58.06 ± 0.21** | 57.66 ± 0.20 |
| | AUC | 68.05 ± 0.56 | 59.54 ± 0.22 | 73.02 ± 0.65 | 77.29 ± 0.47 | 74.32 ± 0.16 | 74.92 ± 0.64 | 78.05 ± 0.10 | **78.85 ± 0.28** | 78.83 ± 0.22 |
| 30% | F1 | 45.05 ± 0.90 | 35.36 ± 0.84 | 49.71 ± 0.30 | 53.93 ± 1.13 | 52.27 ± 0.35 | 49.92 ± 0.36 | 52.00 ± 0.47 | 54.16 ± 0.49 | **54.83 ± 0.15** |
| | AUC | 66.89 ± 0.31 | 60.23 ± 0.31 | 71.28 ± 0.10 | 76.50 ± 0.31 | 72.22 ± 0.52 | 72.52 ± 0.29 | 76.14 ± 0.38 | 76.91 ± 0.41 | **77.10 ± 0.30** |
| 40% | F1 | 43.49 ± 0.95 | 32.23 ± 0.61 | 48.06 ± 0.54 | **50.95 ± 0.99** | 49.32 ± 0.17 | 49.36 ± 0.60 | 49.29 ± 0.28 | 50.72 ± 0.37 | 50.66 ± 0.18 |
| | AUC | 65.32 ± 0.59 | 60.09 ± 0.35 | 69.31 ± 0.10 | 73.24 ± 0.76 | 69.85 ± 0.11 | 70.05 ± 0.46 | 72.08 ± 1.27 | 73.63 ± 0.48 | **73.88 ± 0.25** |
| 50% | F1 | 42.75 ± 1.22 | 33.01 ± 0.51 | 47.39 ± 0.16 | 46.69 ± 0.63 | 43.08 ± 0.47 | 44.44 ± 0.36 | 46.19 ± 0.34 | 47.31 ± 0.20 | **48.13 ± 0.33** |
| | AUC | 62.32 ± 0.42 | 58.20 ± 0.17 | 66.61 ± 0.22 | 70.90 ± 0.82 | 67.63 ± 0.55 | 67.44 ± 0.16 | 70.15 ± 1.10 | **71.74 ± 0.14** | 70.48 ± 0.11 |

*Table 10.* Experiment setting III: Main results on CMU-MOSI

| Test Modality | | | Dataset: CMU-MOSI / Evaluation Metric: AUC | | | | | | | | |
|---|---|---|---|---|---|---|---|---|---|---|---|
| Video | Text | Audio | ShaSpec | mmFormer | TF | LlMoE | FuseMoE-S | FuseMoE-L | FlexMoE | **ConfSMoE-T** | **ConfSMoE-E** |
| ✓ | ✓ | | 62.18 ± 0.88 | 72.72 ± 0.08 | 59.37 ± 1.12 | 74.63 ± 0.13 | 55.35 ± 1.11 | 56.44 ± 1.48 | 74.59 ± 1.50 | **75.57 ± 0.55** | 75.56 ± 0.33 |
| ✓ | | ✓ | 55.79 ± 2.53 | 52.21 ± 0.31 | 57.87 ± 0.32 | 56.84 ± 0.34 | 56.24 ± 1.96 | 55.11 ± 0.77 | 52.28 ± 1.23 | 58.42 ± 0.14 | **58.66 ± 0.20** |
| | ✓ | ✓ | 65.22 ± 1.01 | 71.96 ± 0.47 | 57.89 ± 0.65 | 75.59 ± 0.39 | 54.36 ± 1.67 | 56.10 ± 1.76 | 75.55 ± 1.64 | **77.02 ± 0.63** | 76.96 ± 0.64 |
| ✓ | | | 55.10 ± 1.59 | 71.01 ± 0.53 | 58.57 ± 0.44 | 74.73 ± 0.51 | 55.94 ± 1.51 | 55.66 ± 1.66 | 74.35 ± 1.60 | 75.07 ± 0.70 | **75.11 ± 0.54** |
| | ✓ | | 61.60 ± 4.80 | 73.77 ± 0.44 | 60.28 ± 1.42 | **75.73 ± 0.33** | 55.45 ± 0.95 | 56.58 ± 1.29 | 74.42 ± 1.39 | 75.58 ± 0.55 | 75.50 ± 0.41 |
| | | ✓ | 66.85 ± 3.01 | 69.04 ± 0.94 | 57.19 ± 0.72 | 73.29 ± 0.53 | 56.04 ± 1.91 | 57.85 ± 0.74 | 75.34 ± 1.46 | **77.02 ± 0.63** | 76.47 ± 0.63 |

*Table 11.* Experiment setting III: Main results on CMU-MOSI

| Test Modality | | | Dataset: CMU-MOSI / Evaluation Metric: F1 | | | | | | | | |
|---|---|---|---|---|---|---|---|---|---|---|---|
| Video | Text | Audio | ShaSpec | mmFormer | TF | LlMoE | FuseMoE-S | FuseMoE-L | FlexMoE | **ConfSMoE-T** | **ConfSMoE-E** |
| ✓ | ✓ | | 41.27 ± 0.81 | 48.25 ± 0.11 | 34.96 ± 0.72 | 44.45 ± 0.33 | 30.51 ± 2.53 | 29.90 ± 2.58 | 50.54 ± 0.73 | **54.07 ± 0.29** | 54.04 ± 0.36 |
| ✓ | | ✓ | 36.99 ± 2.03 | 35.79 ± 0.55 | 34.83 ± 0.54 | 35.20 ± 0.64 | 24.29 ± 4.23 | 27.89 ± 6.23 | 30.02 ± 7.82 | 37.40 ± 0.16 | **38.17 ± 0.13** |
| | ✓ | ✓ | 43.72 ± 1.64 | 45.53 ± 0.33 | 37.25 ± 0.53 | 47.79 ± 0.46 | 20.68 ± 2.14 | 29.88 ± 4.55 | 50.41 ± 1.55 | **52.86 ± 0.75** | 51.13 ± 0.15 |
| ✓ | | | 42.98 ± 0.21 | 46.49 ± 0.91 | 38.11 ± 0.35 | 44.86 ± 0.94 | 25.67 ± 2.82 | 27.34 ± 2.94 | 50.39 ± 0.87 | 53.76 ± 0.43 | **54.57 ± 0.35** |
| | ✓ | | 41.36 ± 2.27 | 47.13 ± 0.54 | 37.99 ± 0.40 | 43.81 ± 0.42 | 25.70 ± 2.82 | 27.30 ± 2.95 | 50.65 ± 0.90 | **53.77 ± 0.45** | 49.93 ± 0.06 |
| | | ✓ | 45.45 ± 2.73 | 46.02 ± 1.21 | 38.79 ± 0.56 | 45.50 ± 0.47 | 25.23 ± 4.59 | 29.73 ± 3.51 | 50.57 ± 1.35 | **53.72 ± 0.94** | 50.69 ± 0.47 |

*Table 12.* Experiment setting III: Main results on CMU-MOSEI

| Test Modality | | | Dataset: CMU-MOSEI / Evaluation Metric: AUC | | | | | | | | |
|---|---|---|---|---|---|---|---|---|---|---|---|
| Video | Text | Audio | ShaSpec | mmFormer | TF | LlMoE | FuseMoE-S | FuseMoE-L | FlexMoE | **ConfSMoE-T** | **ConfSMoE-E** |
| ✓ | ✓ | | 64.47 ± 0.10 | 76.60 ± 0.39 | 56.98 ± 0.20 | 81.23 ± 0.07 | 75.81 ± 0.82 | 76.62 ± 0.37 | 80.63 ± 0.41 | 81.45 ± 0.02 | **81.52 ± 0.07** |
| ✓ | | ✓ | 58.41 ± 0.10 | 56.72 ± 0.84 | 58.41 ± 0.17 | 59.64 ± 0.56 | 57.32 ± 0.62 | 57.22 ± 0.12 | 58.28 ± 1.51 | **60.33 ± 0.03** | 60.00 ± 0.23 |
| | ✓ | ✓ | 64.46 ± 0.14 | 75.91 ± 0.04 | 58.64 ± 0.13 | 80.00 ± 0.29 | 77.24 ± 0.60 | 77.31 ± 0.16 | 80.00 ± 0.10 | **81.26 ± 0.10** | 81.23 ± 0.14 |
| ✓ | | | 63.52 ± 0.19 | 75.82 ± 0.28 | 58.71 ± 0.35 | 80.69 ± 0.34 | 76.84 ± 0.14 | 76.27 ± 0.70 | 81.06 ± 0.20 | **81.60 ± 0.13** | 81.53 ± 0.22 |
| | ✓ | | 64.40 ± 0.24 | 75.83 ± 0.28 | 58.89 ± 0.40 | 80.96 ± 0.26 | 76.85 ± 0.14 | 76.51 ± 0.50 | 80.48 ± 0.21 | 81.49 ± 0.13 | **81.65 ± 0.17** |
| | | ✓ | 65.16 ± 0.04 | 75.85 ± 0.09 | 58.32 ± 0.04 | 80.19 ± 0.11 | 77.13 ± 0.75 | 77.23 ± 0.24 | 80.21 ± 0.11 | **81.25 ± 0.10** | 81.23 ± 0.14 |

*Table 13.* Experiment setting III: Main results on CMU-MOSEI

| Test Modality | | | Dataset: CMU-MOSEI / Evaluation Metric: F1 | | | | | | | | |
|---|---|---|---|---|---|---|---|---|---|---|---|
| Video | Text | Audio | ShaSpec | mmFormer | TF | LlMoE | FuseMoE-S | FuseMoE-L | FlexMoE | **ConfSMoE-T** | **ConfSMoE-E** |
| ✓ | ✓ | | 40.75 ± 0.14 | 56.02 ± 0.56 | 31.38 ± 0.61 | 58.97 ± 0.53 | 55.85 ± 0.69 | 53.59 ± 1.07 | 60.10 ± 0.76 | 61.56 ± 0.22 | **61.78 ± 0.58** |
| ✓ | | ✓ | 38.15 ± 0.92 | 37.12 ± 0.66 | 31.73 ± 0.16 | 38.26 ± 2.40 | 29.40 ± 1.67 | 33.29 ± 2.89 | 36.28 ± 3.51 | **39.76 ± 0.23** | 39.36 ± 1.45 |
| | ✓ | ✓ | 40.82 ± 0.12 | 56.06 ± 0.29 | 31.86 ± 0.23 | 58.58 ± 0.31 | 57.02 ± 1.60 | 53.38 ± 0.65 | 58.70 ± 2.10 | **61.37 ± 0.28** | 61.15 ± 0.39 |
| ✓ | | | 40.37 ± 0.46 | 55.70 ± 0.35 | 31.59 ± 0.23 | 61.20 ± 0.46 | 54.02 ± 2.88 | 53.08 ± 0.40 | 58.58 ± 0.71 | **61.84 ± 0.70** | 61.78 ± 0.58 |
| | ✓ | | 42.16 ± 0.22 | 55.89 ± 0.37 | 30.84 ± 0.26 | 59.35 ± 0.12 | 55.87 ± 0.68 | 54.11 ± 0.49 | 58.42 ± 0.27 | 61.56 ± 0.61 | **61.94 ± 0.37** |
| | | ✓ | 42.45 ± 0.85 | 56.06 ± 0.29 | 30.71 ± 0.71 | 60.00 ± 0.85 | 56.64 ± 1.65 | 52.92 ± 1.11 | 59.50 ± 0.80 | **61.10 ± 0.10** | 61.00 ± 0.76 |

## G.1. Additional Ablation Study

Table 14. Control Complexity Study: We control the complexity of model to evaluate performance

| Scale | Metrics | ShaSpec | TF | LlMoE | mmFormer | FuseMoE | FlexMoE | ConfSMoE-T |
|---|---|---|---|---|---|---|---|---|
| Small | MFLOPs | 21,73 | 4.20 | 1245.03 | 237.39 | 143.93 | 40.32 | 39,62 |
| | #Params | 2,177,406 | 2,101,571 | 2,130,227 | 2,378,371 | 20,971,020 | 2,184,963 | 2,184,730 |
| | F1 score | $39.12 \pm 1.21$ | $35.81 \pm 0.85$ | $41.11 \pm 0.90$ | $41.14 \pm 0.31$ | $36.52 \pm 0.84$ | $42.12 \pm 1.18$ | $44.63 \pm 1.65$ |
| Medium | MFLOPs | 111.27 | 45.28 | 6803,84 | 2830.37 | 286.02 | 263.94 | 268.62 |
| | #Params | 11,137,046 | 11,320,153 | 11,469,437 | 11,490,403 | 11,524,392 | 11,371,473 | 11,330,062 |
| | F1 score | $41.56 \pm 0.67$ | $38.62 \pm 0.38$ | $43.21 \pm 0.56$ | $42.44 \pm 0.79$ | $39.44 \pm 0.81$ | $43.78 \pm 0.36$ | $45.37 \pm 1.73$ |
| Large | MFLOPs | 331.87 | 133.47 | 19668.02 | 9282,03 | 308.14 | 1181.48 | 1078,82 |
| | #Params | 33,203,156 | 33,368,335 | 33,000,827 | 33,428,969 | 33,182,556 | 33,385,803 | 33,398,654 |
| | F1 score | $42.82 \pm 0.74$ | $37.64 \pm 0.88$ | $43.45 \pm 0.82$ | $41.50 \pm 0.66$ | $40.84 \pm 0.67$ | $43.46 \pm 1.36$ | $45.55 \pm 1.69$ |

Table 15. Ablation II: Module dropout. CMU-MOSI adopt 50% missing in Experiment II

| Variants | MIMIC-III | | CMU-MOSI | |
|---|---|---|---|---|
| | F1 | AUC | F1 | AUC |
| w/o Conf | $48.59 \pm 1.97$ | $85.91 \pm 0.84$ | $42.47 \pm 0.94$ | $68.77 \pm 1.66$ |
| w/o impute | $49.59 \pm 0.29$ | $86.31 \pm 0.12$ | $42.46 \pm 2.29$ | $68.44 \pm 1.82$ |
| w/o post-impute | $49.86 \pm 1.89$ | $86.53 \pm 0.10$ | $43.14 \pm 1.32$ | $69.13 \pm 1.45$ |
| w/o impute & Conf | $46.32 \pm 1.66$ | $85.15 \pm 0.57$ | $41.98 \pm 0.74$ | $68.48 \pm 1.90$ |
| **ConfSMoE** | $53.44 \pm 0.27$ | $87.05 \pm 0.58$ | $44.34 \pm 0.77$ | $70.41 \pm 1.13$ |

**Ablation III:** We evaluate ConfNet in supervised learning as it is one of the most foundational learning paradiam. For sure, we can adapt ConfSMoE to unsupervised learning setting with proper constructed unsupervied learning loss or proxy task. A typical example is contrastive learning. The supervision signal will be confidence of anchor to it's positive samples. In multimodal setting, contrasitve loss can be computed as the confidence score between two modality in one instance. But this can be combinatorial complexity when the number of modality grows. Specifically, given an instance containing multimodal data $M = \{M_0, M_1, \cdots, M_b\}$. The contrastive loss of any arbitary modality $M_i$ is defined as:

$$\mathcal{L} = -log \frac{e^{M_i \cdot \hat{M}_i / T}}{\sum_{i \neq j}^{B} e^{M_i \cdot \hat{M}_j / T}} \tag{20}$$

Where, $\hat{M}_i$ is interacted modality representation of other modadlity data in one instance. In our setting, $\hat{M}_i$ is the mean representation of other modality. The negative samples are the interacted modality representation of other instances. We first pre-train ConSMoE with contrastive loss stated above and finetune a 3 layer MLP for performance evaluation as it is a common setting for constrastive learning.

As shown in Table 16, we evaluate the unsupervised variant ConfNet on MIMIC-III-48IHM task and CMU-MOSI with 50% missingness. It is clear that the supervised learning setting does not outperform ConfSMoE under supervised learning setting (ConfSMoE-SuP) since supervised learning usually wins well due to the direct supervision signal for a relative small dataset. However, ConfSMoE-UnS significantly outperform FuseMoE with softmax gate (FuseMoE-S), ShaSpec, TF, and achieve comparable performance to LlMoE, mmFormer and FlexMoE under supervised learning setting. This experiment

Table 16. Ablation III: Verification of ConfNet on Unsupervised Learning Setting

| Dataset | Metric | ShaSpec | TF | mmFormer | LlMoE | FuseMoE-S | FuseMoE-L | FlexMoE | ConfSMoE-UnS | ConfSMoE-SuP |
|---|---|---|---|---|---|---|---|---|---|---|
| MIMIC-III | F1 | $34.36 \pm 0.73$ | $43.80 \pm 0.67$ | $23.09 \pm 1.13$ | $47.62 \pm 0.21$ | $48.62 \pm 0.27$ | $51.63 \pm 0.39$ | $47.81 \pm 0.31$ | $51.70 \pm 0.63$ | $53.44 \pm 0.27$ |
| | AUC | $79.86 \pm 0.53$ | $81.27 \pm 0.76$ | $78.47 \pm 0.65$ | $83.30 \pm 0.64$ | $85.81 \pm 0.13$ | $86.22 \pm 0.15$ | $85.10 \pm 0.61$ | $86.88 \pm 0.88$ | $87.05 \pm 0.58$ |
| CMU-MOSI-50% | F1 | $36.39 \pm 0.97$ | $38.46 \pm 1.09$ | $41.14 \pm 0.31$ | $41.54 \pm 0.77$ | $38.55 \pm 0.34$ | $40.90 \pm 0.49$ | $41.95 \pm 0.48$ | $40.78 \pm 0.39$ | $44.34 \pm 0.77$ |
| | AUC | $55.60 \pm 0.12$ | $60.49 \pm 0.80$ | $61.87 \pm 0.67$ | $67.43 \pm 0.34$ | $62.28 \pm 0.28$ | $65.73 \pm 0.30$ | $66.47 \pm 0.49$ | $67.48 \pm 0.15$ | $70.41 \pm 1.31$ |

*Table 17.* Ablation IV: Sensitivity Analysis on Sparsity Hyperparameter (B) in Post-imputation

| Metric | $B=1$ | $B=2$ | $B=3$ | $B=4$ | $B=5$ | $B=6$ | $B=7$ |
|---|---|---|---|---|---|---|---|
| F1 | $44.71 \pm 1.72$ | $43.37 \pm 1.41$ | $43.13 \pm 0.32$ | $44.34 \pm 0.77$ | $43.33 \pm 0.48$ | $43.51 \pm 0.71$ | $43.19 \pm 0.19$ |
| AUC | $69.33 \pm 0.97$ | $68.45 \pm 0.57$ | $69.45 \pm 1.60$ | $70.41 \pm 1.31$ | $69.77 \pm 1.31$ | $68.92 \pm 1.15$ | $68.12 \pm 0.23$ |

should be sufficient to show that ConfSMoE can be extended to unsupervised learning scenario. We also highlight that the performance may be impacted by the proxy task as it dirrect impact the quality of representation.

**Ablation IV:** The sparsity hyperparameter in post-imputation determines the amount of feature retrieved from available modality and filtering out irrelavant feature from them. The higher the sparsity $B$ is, the sparser the attention mechanism will be. The post-imputation module will be standard attention if $B = 1$. In the paper, we keep $B = 4$ as this is the best parameter we search in range [1, 7]. To show the sensitivity of $B$, we implemented experiment to evaluate how $B$ hyperparameter impact the model performance as shown in Table 17

*Table 18.* Ablation V: Sensitivity Analysis on Calibration of Confidence Supervision

| Metric | Temp=0.3 | Temp=0.6 | Temp=1 | Temp=2 | Temp=3 |
|---|---|---|---|---|---|
| F1 | $43.62 \pm 1.07$ | $43.77 \pm 0.94$ | $44.35 \pm 0.77$ | $44.18 \pm 0.11$ | $43.08 \pm 0.37$ |
| AUC | $69.75 \pm 1.34$ | $68.74 \pm 0.38$ | $70.41 \pm 1.31$ | $71.14 \pm 0.61$ | $69.22 \pm 0.6$ |
| ECE(%) | $10.85 \pm 1.72$ | $8.61 \pm 1.27$ | $8.32 \pm 1.27$ | $6.34 \pm 2.80$ | $9.01 \pm 0.96$ |

**Ablation V:** As shown in Table. 18, by changing the temperature parameter of softmax in downstream task head, we can easily control the calibration of model. We use a wildly used metrics Expected Calibration Error (ECE) to measure how well the model is calibrated to actual accuracy. The ECE is closer to 0, the better the calibration is. Specifically, we set the temperature as 1 to indicate normal softmax, and large temperature value to make over-calibrated model and small temperature to make under-calibrated model. Both cases are not well-calibrated. We can clearly observed that temp $= 2$ is the best calibration setting and it is also the best performance in terms of AUC and comparable performance in F1 score. we can observed that worse calibrated confidence is definitely harmful to ConfNet as it is a true reflection of the instance contribution to downstream task. But the model performance is not very sensitive to calibration of model. A clearer trend is that better calibration may lead to better performance and vice versa.

### G.2. Further Clarification

**Connection of Both ConfNet and Tow-Stage Imputation**: Conceptually, when a modality is missing, the input signal to a standard softmax-based gating network is partial, producing unreliable representations for gating score learning (Eq. 3). Traditional softmax gating always produces high gating scores for the selected expert due to the natural exponential function of softmax even if the input signal is unreliable. This is counterintuitive: an unreliable modality representation receives a high gating score. The model will propagate this error during training: it always produces strong gating scores no matter the quality of the modality representation. This leads to worse expert collapse and thus worse performance and unreliable routing.

ConfNet gating mechanism, however, measures the input signal's downstream task confidence. When the input signal is unreliable, the confidence score will be relatively small for the downstream task. We take this low confidence score as the gating score to select experts such that the output of the selected expert will contribute less to the final representation from this unreliable signal. Therefore, the router will tend to select other experts for better representation learning for the next round, leading to a natural balance of expert load. Our two-modality imputation here is to reconstruct the missing modality to guide the routing process with the confidence score to learn better gating scores w.r.t. the quality of representation.

In addition, the key characteristic of post-imputation is to integrate the "opinion" from different experts, which brings better imputation capability than a single model and this is also the key difference compared to other single-model imputation.

# H. Visualization of Experiment

In this section, we provide visual evidence to further support our claims regarding expert selection dynamics and modality robustness.

Figure 4 shows the UMAP projection of multi-modal embeddings on CMU-MOSI. It clearly demonstrates that imputed modalities form distributions consistent with their original counterparts, suggesting that our imputation module maintains semantic alignment across modalities.

Figure 5 illustrates the softmax gating mechanism without load balance regularization, which exhibits severe expert collapse where only a few experts are consistently selected throughout training. In contrast, Figure 6 shows that our proposed ConfNet gating maintains both load balance and expert specialization, effectively mitigating collapse and enabling more stable and diverse expert routing. We further compared the visualization of our proposed method with Figure 7. SoftMax gating with load balance loss indicate severe ambiguous selection, experts that were heavily selected in one epoch tend to be suppressed in the next epoch. This is a clear empirical result to our gradient analysis that the optimization process of load balance is opposite to learning better routing score. This ambiguous selection, "Sinusoidal Wave" in another word, is also observed in Mean and Gaussian gating as shown in Figure 9 and Figure 11. Mean gating considers all experts equally important and failed to reflect the specification of each expert, resulting suboptimal performance shown in Table 2. By comparing Figure 7 and Figure 9, 11, we observe softmax and gaussian exhibits similar "Sinusoidal Wave" to Mean selection. We can draw a preliminary conclusion that this ambiguous selection, usually leads to suboptimal solution given empirical facts in 2. Moreover, although Laplacian and ConfNet gating are not fully balanced, the "Sinusoidal Wave" exists only to a mild extent and lead to a worth noting performance in 2. All these empirical findings are consistent with our gradient analysis and further support our claim that load distribution should preserve specialization while ensuring a non-trivial load is assigned to less active experts.

Finally, Figures 12 and 13 report and visualize Experiment III results. ConfMoE demonstrates superior resilience across increasing missing rates, with both F1 and AUC metrics degrading gracefully. These results visually confirm the effectiveness of confidence-guided gating and modality imputation under challenging conditions.

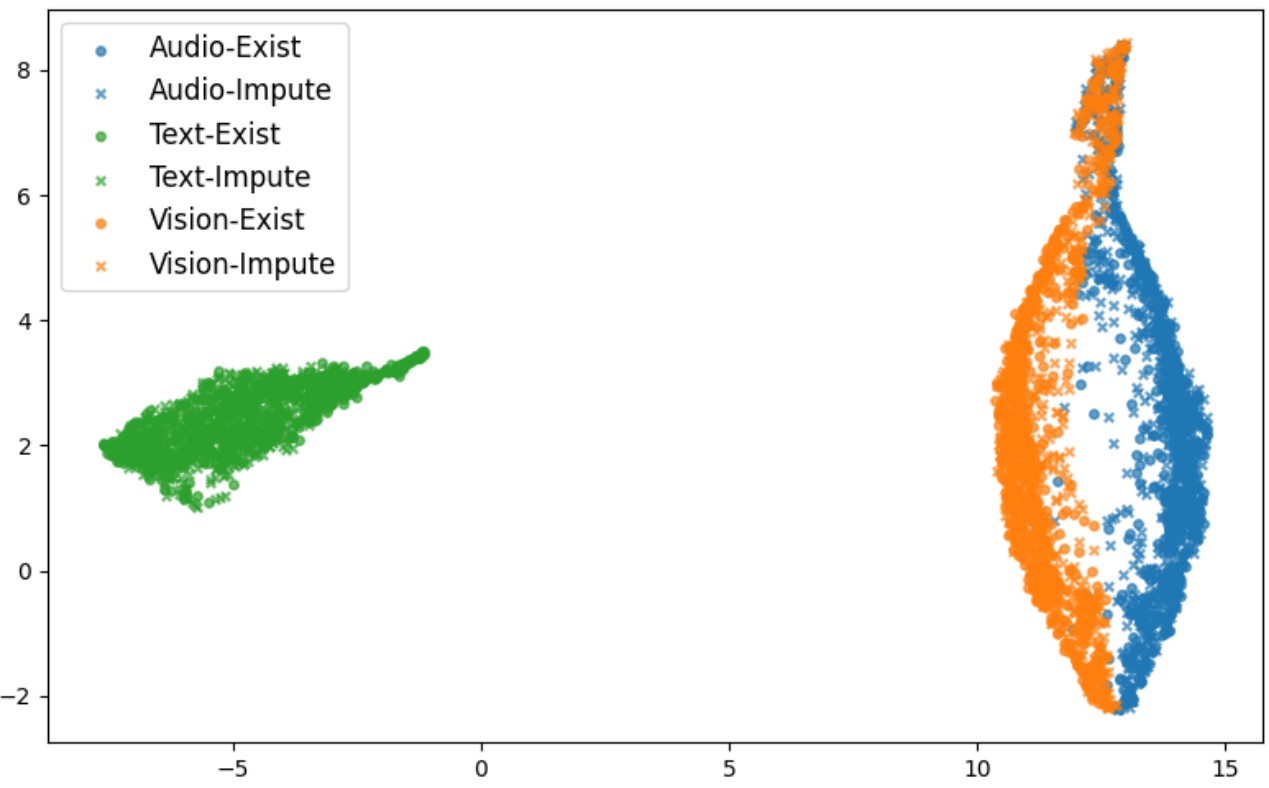

*Figure 4.* UMAP Visualization of Multi-modal Embedding Space: CMU-MOSI

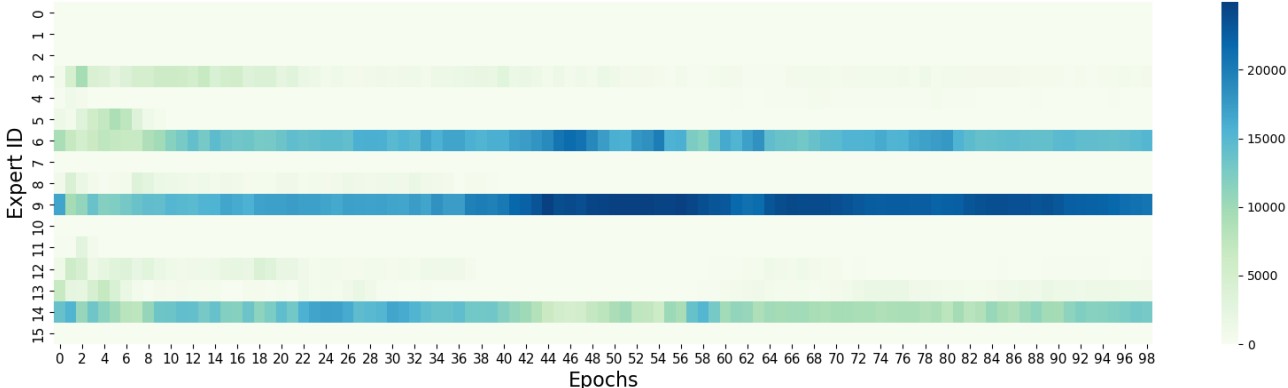

*Figure 5.* Softmax Score-based without Load Balance Expert Selection: Expert collapse is consistently occurred after epochs 10. While a few certain experts are preferred, this preferred selection becomes worst when the training progresses

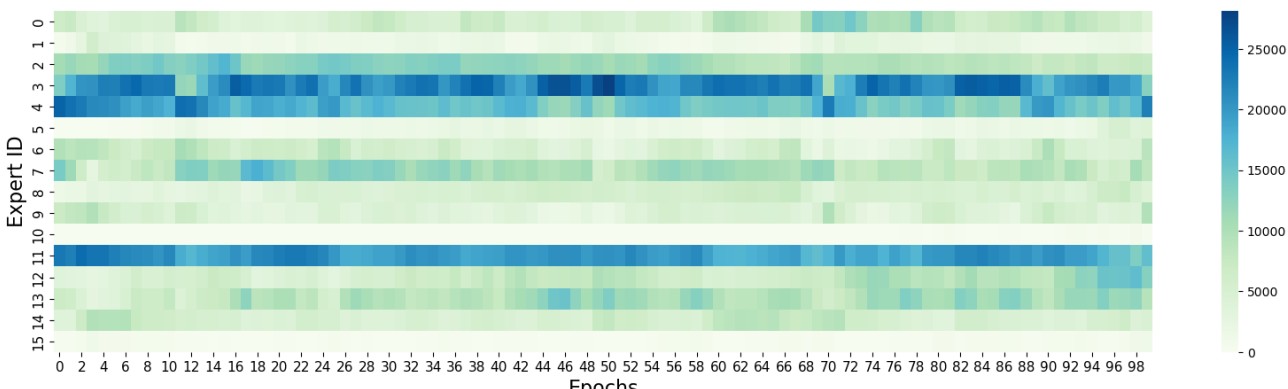

*Figure 6.* ConfNet Gating Expert Selection: Expert collapse is consistently mitigated throughout training. While certain experts are preferred, less frequently selected experts are still actively utilized. Moreover, expert selection becomes increasingly balanced as training progresses, indicating improved stability and diversity in routing.

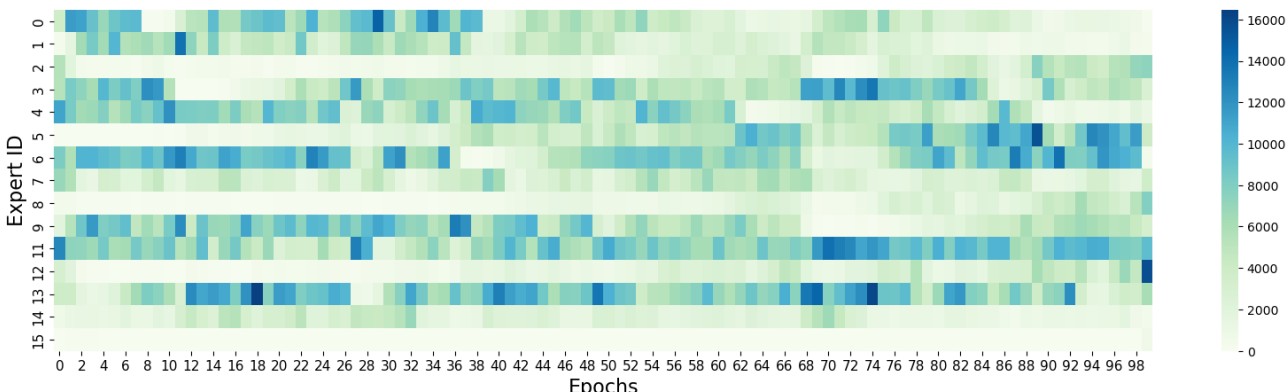

*Figure 7.* Softmax Gate with Load Balance Loss Expert Selection: An optimization conflict arises, often reflected in a "Sharp Sinusoidal Wave" pattern, experts that were heavily selected in one epoch tend to be suppressed in the next. This oscillatory behavior indicates that the typical load balancing loss does not promote true specialization or expertise among experts. Instead, it merely enforces uniform usage, potentially at the cost of performance, by prioritizing balanced selection over actual expert quality.

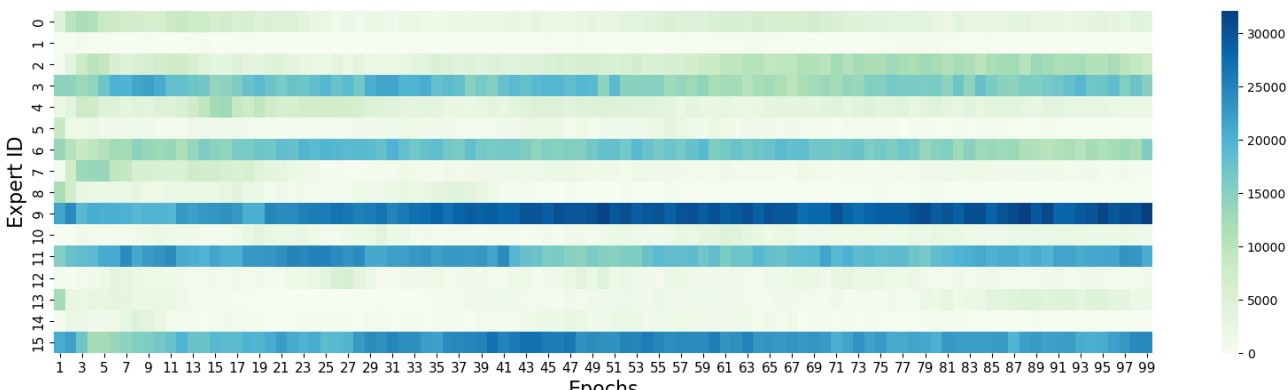

*Figure 8.* Laplacian Gating Expert Selection: Laplacian gating can partially alleviate the load imbalance issue and achieves better expert utilization than softmax. However, as training progresses, selection becomes increasingly skewed, with only a few experts being consistently chosen. This eventually leads to expert collapse, where most experts become inactive and underutilized.

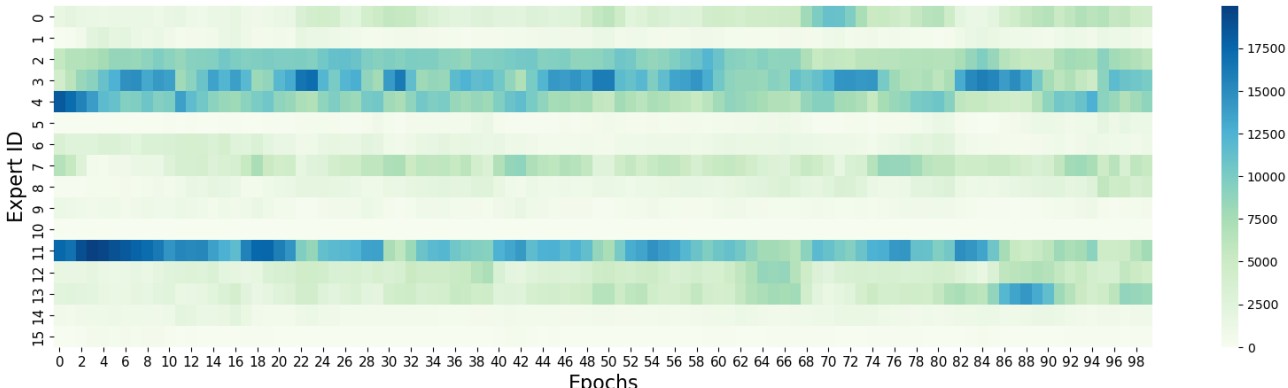

*Figure 9.* Mean Gating Expert Selection: The "Sharp Sinusoidal Wave" pattern persists under mean gating, where experts heavily selected in one epoch are suppressed in the next. This oscillatory behavior reflects ambiguity in expert selection, stemming from the absence of explicit guidance in the routing mechanism. While mean gating achieves better load balance than Softmax with $\mathcal{L}_{\text{load}}$, it ultimately results in inferior performance, as shown in Table 2.

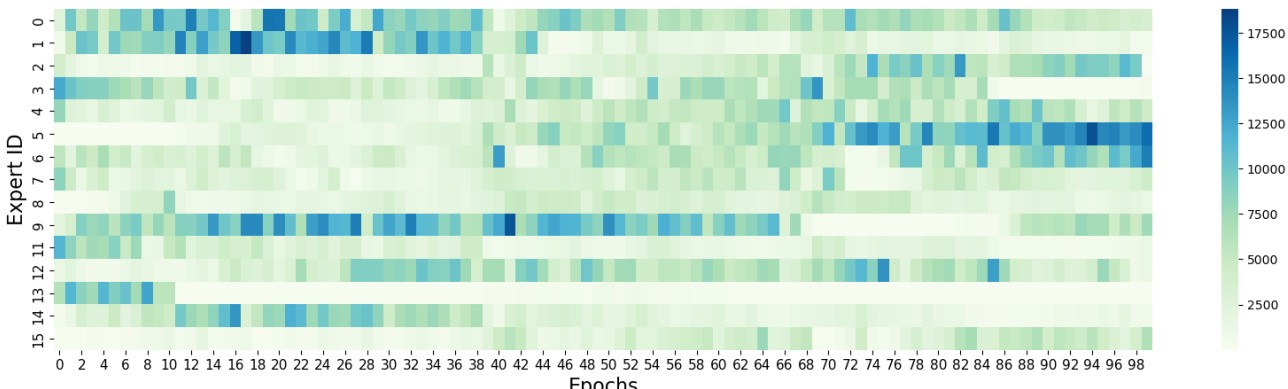

*Figure 10.* Gaussian Gating Expert Selection: The "Sharp Sinusoidal Wave" pattern persists under Gaussian gating, where experts heavily selected in one epoch are suppressed in the next. This oscillatory behavior reflects ambiguity in expert selection, stemming from the inappropriate gating score guided from Gaussian. While mean gating achieves better load balance than Softmax with $\mathcal{L}_{\text{load}}$, it ultimately results in inferior performance, as shown in Table 2.

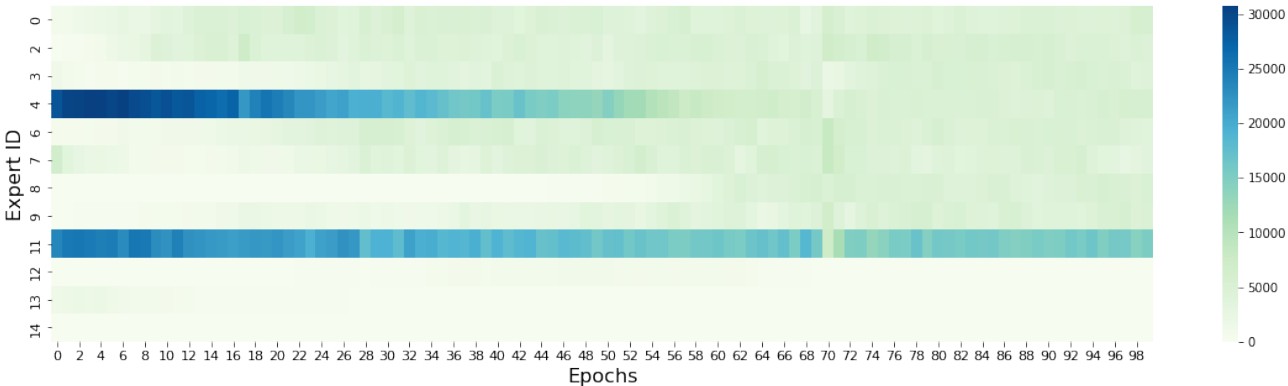

*Figure 11.* Loss-Free Balance Expert Selection: The "Sharp Sinusoidal Wave" pattern is mitigated under Loss-Free Balance. But this balance technique works too slow as it start balancing load since epoch 50.

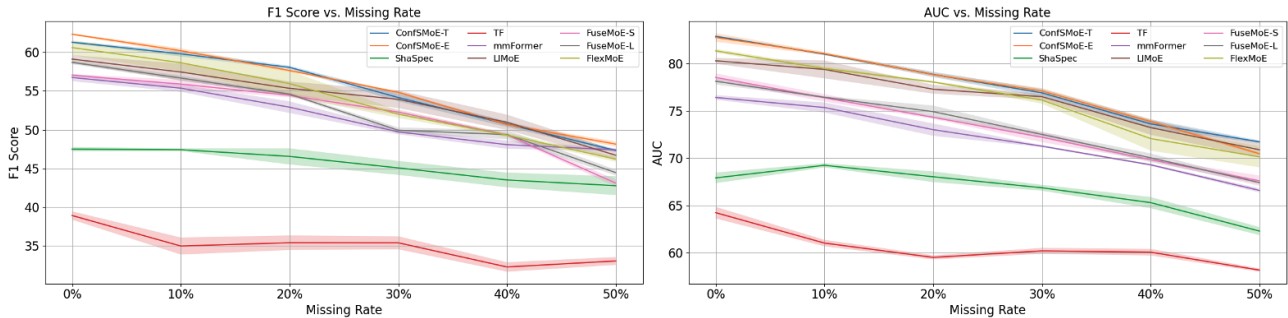

*Figure 12.* Experiment III: CMU-MOSEI Performance Plot

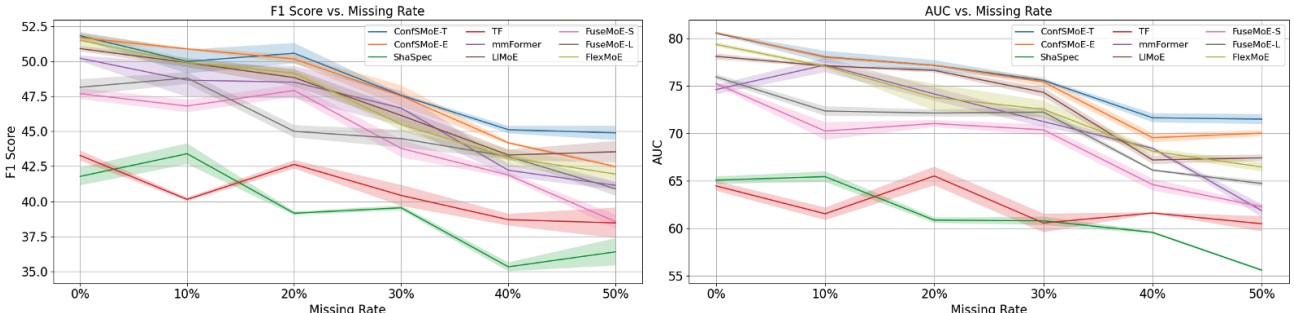

*Figure 13.* Experiment III: CMU-MOSI Performance Plot

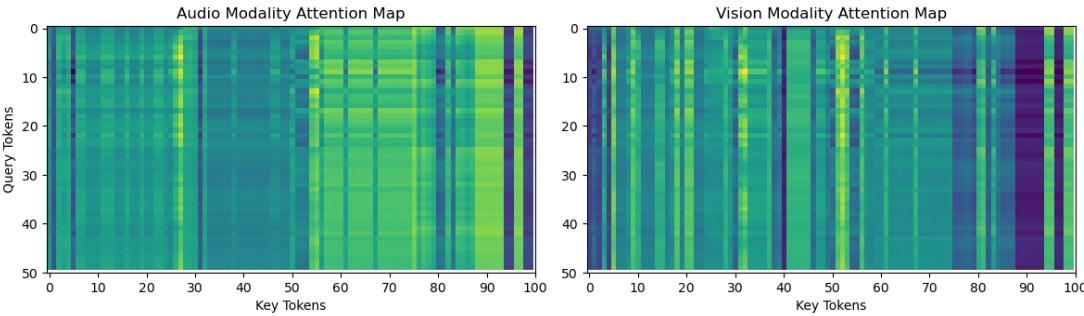

*Figure 14.* Attention Map from post-imputation: Text as query and Vision, Audio as key. The key tokens are obtained by concatenating tokens from K=2 expert, resulting 100 length of token. The darker the color, the lower attention weight is.

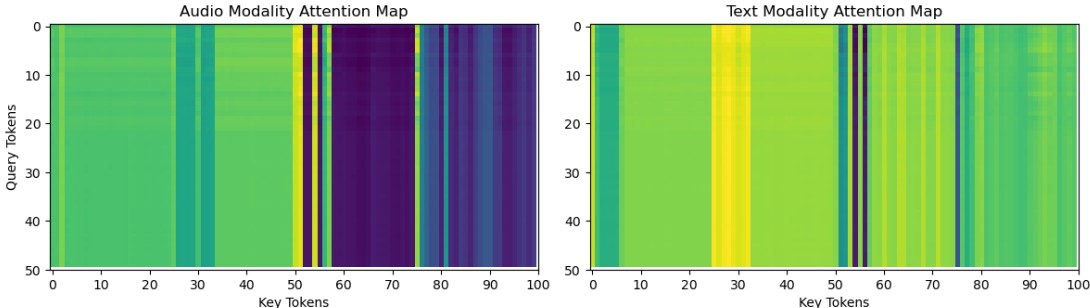

*Figure 15.* Attention Map from post-imputation: Vision as query and Audio, Text as key. The key tokens are obtained by concatenating tokens from K=2 expert, resulting 100 length of token. The darker the color, the lower attention weight is.

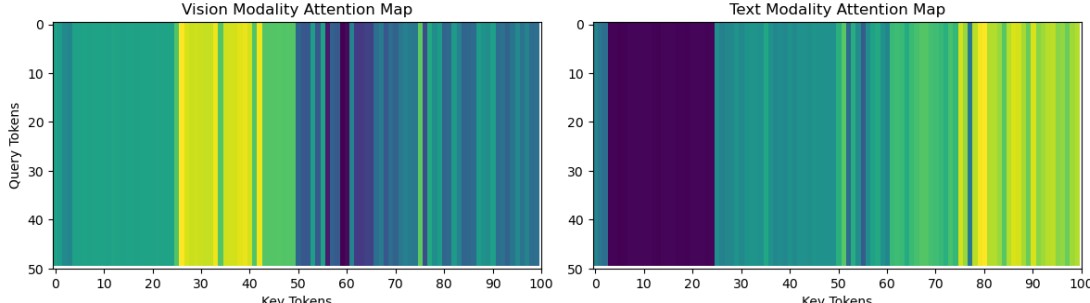

*Figure 16.* Attention Map from post-imputation: Audio as query and Vision, Text as key. The key tokens are obtained by concatenating tokens from K=2 expert, resulting 100 length of token. The darker the color, the lower attention weight is.

From Figure 14 15 16, we can clearly observed that the attention map are distinctly different when the key are set to different modalities. This indicate that the post-imputation module can dynamically learn, which token are needed depending on characteristics of token and experts. For example, the Figure 16 indicates that 25 - 45th tokens of the first expert are more important than others while the text modalities shows that the second experts are more important. Similar observation are also shows in 14 15 as well. This observation also indicates that modality not only dynamically select information from existing modality but also the specialization of experts. Therefore, we can conclude that the post-imputation can capture expertise of different expert and refine the missing modality accordingly.

