# OpenReview forum: "Rethinking Gating Mechanism in Sparse MoE: Handling Arbitrary Modality Inputs with Confidence-Guided Gate"
_ICML.cc/2026/Conference — ICML 2026 regular_

### Official Review · Reviewer_5Krs · 2026-02-28

**Soundness:** 3
**Presentation:** 2
**Significance:** 3
**Originality:** 3
**Overall Recommendation:** 4
**Confidence:** 2

**Summary:**

This paper proposes the ConfSMoE framework, a solution for modal missing tasks based on sparse expert mixture models. Key contributions include: (i) ConfNet: a Confidence-Guided Gating Network, and (ii) a two-stage framework for completion and cross-modal contextual refinement. Through gradient analysis, this paper investigates expert collapse phenomena under softmax routing and load-balance losses, and validates model performance across multiple datasets in diverse modality-dropout scenarios.

**Compliance With Llm Reviewing Policy:**

Affirmed.

**Final Justification:**

The authors addressed my concerns. However, since I am not an expert in this field, I decided to keep my original positive rating (4).

**Key Questions For Authors:**

1.    Is $c_i$ normalized among the selected experts? If not normalized, how is the output scale controlled to avoid instability and ensure fair competition among experts?

2.	Please clarify the difference between the theoretical analysis based on $1/H(g)$ and the actual load balancing loss (in CV$^2$ form) used in the experiment. Under the CV$^2$ formula, do your conclusions regarding gradient conflicts still hold?


Overall, this paper offers an interesting solution for the modal missing task, with encouraging empirical results and a research approach of potential value. However, the manuscript's writing and formatting require further refinement, though this does not diminish the paper's contributions as these issues can be effectively addressed. Nevertheless, given that I am not an expert in this field and lack deep background understanding, the AC should adjust the weighting of my comments as appropriate.

**Limitations:**

see questions

**Strengths And Weaknesses:**

**Strength:**

1. Handling missing modal data and preventing expert model collapse is one of the key challenges in multimodal SMoE architectures.
2. Decoupling expert aggregation weights from softmax routing through supervised confidence signals offers an innovative perspective for addressing expert collapse.

**Weakness:**

1. The paper's writing is somewhat disorganized, obscuring the core message the authors aim to convey. For instance, in the Introduction section, the opening of the second paragraph asserts that the expert collapse issue in SMoE has been resolved, yet handling modality missing cases remains an open problem. Subsequent sections then shift focus to introducing primary strategies for addressing modal missingness, decoupling from the earlier SMoE narrative. Furthermore, multiple research motivations are interwoven throughout the lengthy text, diminishing readability.

2.    The article's formatting is poor, though this does not significantly undermine its contributions.

---

> ### Author Rebuttal · Authors · 2026-03-27
>
> # Response to Weaknesses 1 & 2:
>
> We thank the reviewer to identify the readability issue and endorse our contribution and significance. To clarify, we have some revision to build better connections between MoE and the missing modality problem.
>
> Additionally, we have revised the second paragraph to clarify the motivation of our work as follows. To comply with ICML rebuttal policies, we integrated these updates without significantly altering the section's overall structure.
>
>
> >When one or more modalities are missing, many imputation approaches aim to reconstruct absent views through cross-modal information sharing  (Ma et al., 2021; Wang et al., 2023; Yao et al., 2024), as shown in Figure 1b and overlook modality-specific information. We hypothesise that relying solely on available-modality imputation can misdirect the routing mechanism; specifically, because the imputed features are dominated by the signal of available modality, the router becomes biased. For instance, if a text modality is imputed from vision, the router is likely to assign that 'pseudo-text' to a vision expert rather than a text-specific one, exacerbating expert collapse and degrading gating reliability. Therefore, current modality imputation methods degrade the reliability of the gating mechanism and result in suboptimal expert selection. Although existing SMoE approach attempt to bypass imputation by assigning different modality combinations to designated experts (Han et al.,2024), this strategy becomes computationally intractable as the number of modalities grows exponentially. Another recent work adapt a learnable modality bank  (Yun et al., 2024) to impute missing inputs based on the available modality combinations and enforce load balance by load balance loss. However, the imputation bank is randomly initialised, similar to prompt tuning (Jia et al., 2022; Lester et al., 2021), and fails to capture modality-specific and fine-grained instance-specific structure, limiting the reliability of modality imputation and potentially misdirecting the router. Furthermore, enforcing load balancing via auxiliary loss functions can lead to unstable or ambiguous expert selection, as illustrated in Figure 3a.
>
>
>
> # Response to Question 1:
>
> The predicted confidence score $c_i$ is not normalized at the output of ConfNet router, but encouraged to be as close to ground truth confidence, which is normalized in the range [0, 1]. In addition, all feature input to ConfNet for confidence prediction will be passed to a LayerNorm layer for feature normalization. In practice, we did not observe numerical instability in our experiment.
>
>
> # Response to Question 2:
>
> Thank you for the insightful question.
>
> We consider minimizing $1/H(g)$ in gradient analysis as the general form of flat gating distribution, as we observe most load balance loss aims to achieve a fair selection by forcing a flat distribution. To clarify, let's consider the applied load balance loss in our experiment:
>
>
>
>
> $$
> \begin{aligned}
> \mathcal{L}_{\text{balance}} = CV^2 ( \sum_j^{N} importance_j ) + \mathbf{CV}^2 ( \sum_j^{N} load_j )
> \end{aligned}
> $$
>
> $$ \text{importance}\_j = \sum\_{i=1}^{|D|} g\_{i,j}, \quad \text{load}\_j = \sum\_{i=1}^{|D|} \delta(g\_{i,j} > 0) $$
>
>
>
> where $CV^2(x) = \left( \frac{\sigma(x)}{\mu(x)} \right)^2$, $\sigma(x)$ is the standard deviation of $x$,  $importance_j$ is the expert importance of expert $j$ and $load_j$ is load of expert $j$,  $\delta (\cdot > 0)$ is an indicator function that is 1 when the inner value is greater than 0.
>
> Now we show that this load balance function encourages flat gating score distribution. The denominator is the average across all experts, and then we have
>
> $$
> \begin{aligned}
>     \mu = \frac{1}{N} \sum_{j}^{N} importance_j = \frac{1}{N} \sum_{j}^{N} \left( \sum_i^{|D|} g_{i,j} \right) = \frac{1}{N} \sum_i^{|D|} \left(  \sum_{j}^{N}g_{i,j} \right) = \frac{|D|}{N}
> \end{aligned}
> $$
>
>
> Note that $\sum_{j}^{N}g_{i,j}=1$ is the summation of the gating score out of the softmax router. From above, the denominator is a constant only impacted by the number of instances $|D|$ and the number of experts $N$. Thus, the applied load balance loss will minimize the variance of importance for each expert, which encourages a flat gating score distribution. This is equivalent to minimize the reverse of the gating score entropy $\frac{1}{\mathcal{H}(\textbf{g})}$ since the denominator is a fixed number once the model parameter is decided. This derivation obtains the same conclusion as our gradient analysis.
>
> We will revise session E.3 to provide a better explanation

---

> > ### Author Rebuttal · Reviewer_5Krs · 2026-04-01
> >
> > Thank you to all the authors for their careful responses, and I'm also glad that they could agree with my suggestions regarding the manuscript. The authors addressed my concerns. However, since I am not an expert in this field, I have decided to keep my original score and have asked AC to reduce the weight of my rating.

---

> > > ### Author Response · Authors · 2026-04-01
> > >
> > > Thank you once again for your thoughtful review. We are greatly encouraged by your insightful question, as it demonstrates a grasp of the central concepts behind our work. We appreciate the time and effort you have dedicated to reviewing our paper.

---

### Official Review · Reviewer_P388 · 2026-03-12

**Soundness:** 3
**Presentation:** 1
**Significance:** 3
**Originality:** 3
**Overall Recommendation:** 4
**Confidence:** 3

**Summary:**

In this paper, the authors propose ConfSMoE, which replaces softmax gating with a confidence-guided routing mechanism (ConfNet) and introduces a two-stage modality imputation strategy to address the challenge of handling missing modality and the expert collapse issue in SMoE. The proposed method is evaluated on four real-world datasets with three distinct experiment settings, outperforming eight baselines.

**Compliance With Llm Reviewing Policy:**

Affirmed.

**Key Questions For Authors:**

See weaknesses.

**Limitations:**

yes, the authors mentioned limitations in the Appendix.

**Strengths And Weaknesses:**

### Strengths:

* The discovered problem of current SMoE methods under missing modality multimodal learning is well-motivated, with experiments grounded in widely-used benchmarks.

* The proposed ConfNet and the two-stage imputation module are designed to be architecture-agnostic and can in principle be integrated into existing MoE frameworks without major structural modifications.

* The authors evaluate across multiple missing modality settings with varying missing rates, providing a reasonably thorough empirical coverage of the problem.




### Weaknesses:

* Writing. Several typographical errors have been found in the manuscript, which introduce ambiguity about the method. For example, the method is inconsistently referred to "ConfMoE" and "ConfSMoE" in Table 6 vs. the main text. Typo errors like "Sinoual Wave" in line 70, "axillary loss" in line 170 and line 177.

* Lack of Theoretical Justification. ConfNet is trained to predict $c_i$ toward $p_t$ via an MSE loss. However, $p_t$ is also predicted by the model being trained and evolves throughout optimization. In early training stage, when $p_t$ is not well-trained, the confidence scores $c_i$ learned by ConfNet are necessarily unreliable. Could the authors provide detailed training curves or theoretical analysis to demonstrate that the model develops in a reliable direction as the training process.

* Lack of Ablation Study about the Hyperparameters. In Equation 6, the loss function assigns equal weight to $L_{task}$ and $L_{conf}$ across all experimental settings as described in Table 5. There are no ablation over the loss weight across different datasets or discussion about the choice of a certain weight. This omission leaves a meaningful gap in the practical guidance the paper provides.

* Absence of ablation on the pre-imputation sample size. The pre-imputation stage constructs the representation for the missing modality by averaging $n$ randomly sampled instances from the modality pool. The choice of $n$ is quite sensitive but there's no ablation over $n$ reported in the paper. It is unclear whether the observed performance gains are robust to this choice.

---

> ### Author Rebuttal · Authors · 2026-03-27
>
> # Response to weakness 1:
>
> We thank the reviewer for identifying typos and endorsing our contribution and significance. We have now carefully revised the typos you mentioned and fixed additional typos we found. Please see the response to Reviewer o3Di for details revision.
>
>
> # Response to weakness 2:
>
> The design of ConfNet router estimates how well the expert can handle inputs. If the initial confidence is not reliable, then the estimation will not be reliable since all parameters are randomly intialized. However, the initial unreliability facilitates a healthy exploration phase to learn expert specialization. ConfNet aims to predict the task confidence $P(y|x)$, meaning it inherently stabilises as the entire backbone and downstream head converge, a phenomenon empirically validated in our training logs. Additionally, $\mathcal{L}_{conf}$ relies on a convex MSE formulation, and gradient descent is guided by smooth, stable updates that consistently drive the network toward an optimal solution. We will also add a training convergence plot in the appendix to support our claim.
>
> We provide a train-test loss curve of MIMIC-III in an anonymous GitHub: https://anonymous.4open.science/r/Anomynous_repo-DE0E/
>
> This curve indicates that the ConfSMoE training is not difficult to converge.
>
>
> # Response to weakness 3:
>
> We provide experiments on 50% missing CMU-MOSI dataset to study the impacts of the loss weight. We denote the weight of $\mathcal{L}_{conf}$ as $\lambda$.
>
>
>
> We can clearly see that too weak or too strong constraints from $\mathcal{L}_{conf}$ can lower the performance, while $\lambda$=1 is the best parameter we have tried.
>
>
>
> Increasing $\lambda$ from 0 to 1 leads to a consistent improvement in the F1 score (from 41.74 to 44.34). This suggests that stronger enforcement of the confidence-guided routing helps better confidence prediction from ConfNet. This demonstrates that the confidence-guided gate is crucial for handling missing modalities in the ConfSMoE architecture.
>
> | Metric | $\lambda$=0 | $\lambda$=0.2 | $\lambda$=0.5 | $\lambda$=1 | $\lambda$=2 |  $\lambda$=3 |
> | --- | --- | --- | --- | --- | --- | --- |
> | F1  | 41.74 $\pm$ 1.97 | 43.47 $\pm$ 0.93 | 44.06 $\pm$ 1.47 | 44.34 $\pm$ 0.77 | 43.89 $\pm$ 1.22 | 43.76 $\pm$ 1.33 |
> | AUC | 68.77 $\pm$ 1.66 | 68.63 $\pm$ 0.90 | 68.04 $\pm$ 1.66 | 70.41 $\pm$ 1.31 | 69.39 $\pm$ 1.71 | 68.56 $\pm$ 1.49 |
>
>
> We will add this ablation study to our revised manuscript.
>
>
> # Response to weakness 4:
>
> We show the impacts of the number of instances used in pre-imputation. Sampling 20 instances provides the perfect balance. It captures enough representative features of the missing modality to be informative, while maintaining enough stochasticity to keep the imputed representations distinct across different training batches. If $n$ is too small, the pre-imputed feature relies on too few random samples, injecting excessive noise and variance into the network. This makes it difficult for the ConfSMoE architecture to establish a reliable baseline representation before it attempts the sparse cross-attention refinement in the post-imputation stage. The core design philosophy of this pre-imputation strategy is to introduce stochasticity to prevent the imputed features from collapsing into a single, deterministic representation. By the Law of Large Numbers, as $n$ grows excessively large ($n \to 1000$), the sampled mean converges precisely to the global mean of the entire training set. This creates "deterministic and uninformative" representations that strip the model of instance-level diversity and severely limit the router's ability to learn meaningful gating scores.
>
> | Metric | n=10 | n=20 | n=50 | n=100 | n=200 | n=500 | n=1000 |
> | --- | --- | --- | --- | --- | --- | --- | --- |
> | F1  | 44.34 $\pm$ 0.77 | 45.93 $\pm$ 1.30 | 45.43 $\pm$ 1.40 | 44.63 $\pm$ 1.21 | 44.81 $\pm$ 2.03 | 44.08 $\pm$ 2.39 | 43.79 $\pm$ 1.86 |
> | AUC | 70.41 $\pm$ 1.31 | 71.14 $\pm$ 1.03 | 70.71 $\pm$ 1.63 | 69.70 $\pm$ 1.76 | 69.50 $\pm$ 1.02 | 68.22 $\pm$ 1.87 | 67.89 $\pm$ 2.42 |
>
>
>
> We will add this ablation study to our revised manuscript.

---

> > ### Author Rebuttal · Reviewer_P388 · 2026-04-03
> >
> > Thanks for the authors' detailed response. My concerns have been adequately addressed, and I will keep my original score.

---

> > > ### Author Response · Authors · 2026-04-04
> > >
> > > We thank the reviewer for the acknowledgment. As we have addressed all your questions and improved our writing. Would you please consider raising any point based on our responses?
> > >
> > > All the best :D

---

### Official Review · Reviewer_o3Di · 2026-03-17

**Soundness:** 3
**Presentation:** 3
**Significance:** 4
**Originality:** 3
**Overall Recommendation:** 5
**Confidence:** 5

**Summary:**

This paper studies sparse mixture-of-experts for multimodal learning under missing modalities. The authors argue that two issues are entangled in prior SMoE systems: missing-modality robustness and expert collapse. Their proposed method, ConfSMoE, has two main components: a confidence-guided gating network that replaces softmax routing scores with confidence scores supervised by task signals, and a two-stage imputation pipeline that first reconstructs modality-specific structure from a modality pool and then refines it with sparse cross-modal attention using expert outputs. The paper also presents a gradient-based explanation for why softmax routing plus load-balancing losses can create optimization conflict and unstable expert selection.

**Compliance With Llm Reviewing Policy:**

Affirmed.

**Final Justification:**

check mine and Reviewer 5Krs's reply. good work on an important problem. recommend for accept.

**Key Questions For Authors:**

see above

**Limitations:**

yes

**Strengths And Weaknesses:**

Strengths

The paper’s strongest conceptual contribution is the diagnosis of expert collapse as a gradient issue. The authors analyze the Jacobian of softmax-gated MoE and argue that sharp routing distributions create a rich-get-richer feedback loop, while auxiliary load-balancing gradients can conflict with the main routing gradients.

The problem is important. Missing modality is a central obstacle in real-world multimodal learning


Weaknesses and Questions

1. Please consider asking help from native speakers or LLMs to conduct a full grammar check and list what u changed in rebuttal, otherwise I will consider lowering the presentation to poor.

2. The theory is suggestive but not fully convincing. For example, the claim that the objective in Eq. 6 is convex because cross-entropy and MSE are convex is too loose, since the full model contains neural networks and routing decisions, so global convexity does not follow from the loss choice alone.

3. The pre-imputation strategy seems somewhat heuristic. Sampling random instances from the modality pool and averaging them may inject useful modality priors, but it is not obvious that this preserves patient-specific or semantically appropriate structure before the second-stage refinement.

4. Please cite previous and recent works of similar topics and motivation : REMIND: Rethinking Medical High-Modality Learning under Missingness--A Long-Tailed Distribution Perspective

5. Is the router still learning expert specialization, or is it partially learning a proxy for correctness conditioned on labels?

---

> ### Author Rebuttal · Authors · 2026-03-27
>
> # Response to Weakness 1:
>
> We would like to thank the efforts made by Reviewer o3Di to review our work. We have made appropriate modifications to improve our writing and listed changes below:
>
> 1. line 29: Revise "reveal" -> "revealing"
> 2. line36: Revise "aligns" -> "align"
> 3. line 38: Revise "real world dataset" -> "real-world datasets"
> 4. line 70: "Sinoual" -> "Sinusoidal"
> 5. line 71: Revise "reflecting" -> "reflected"
> 6. line 49-81: Revise the second paragraph of the introduction, revise grammar and clear motivation (see responses Reviewer 5Krs for details)
> 7. line 93: Revise "leverage" -> "leverages"
> 8. line 117: Revise "resulting a sparse resulting expert collapse" -> "resulting expert collapse"
> 9. line 169: "shaper" -> "sharper"
> 10. line 170, line 177 "axillary" -> "auxiliary"
> 11. line 186: Revise "produce" -> "produces"
> 12. line 254: Revise "to collapse" -> "from collapsing"
> 13. line 294: Revise "benchmarking task" -> "benchmarking tasks"
> 14. line 300: "tradidtional" -> "traditional"
> 15. line 231: Revise "setting" -> "settings"
> 16. line 317: Revise "impute" -> "imputed"
> 17. line 345: Revise "worse" -> "worst"
> 18. line 364: Revise "keep" -> "kept"
> 19. line 405: Revise "diagnose" -> "diagnosis"
> 20. line 420: Revise "While in many works considers" -> "While many works consider"
> 21. line 423: "incoperates" -> "incorporates"
> 22. Table 6: "ConfNet-T", "ConfNet-E" -> "ConfSNet-T", "ConfSNet-E"
> 23. Appendix E.3: Revise for grammar, and provide more interpretation (see responses in Reviewer 5Krs for details)
> 24. Line 182: Revise "Propose Method" -> "Proposed Method"
>
> # Response to Weakness 2:
>
> Our focus is on the convexity of the individual loss functions rather than the global loss landscape of the neural network, which is inherently non-convex. This loss convexity ensures a predictable optimization behaviour at the output level, avoiding the gradient conflicts often seen in load balance loss.
>
> We have now rephrased the statement in line 182 to:
>
> >Note that while the global optimization landscape of the deep neural network remains non-convex, the selected loss components (Cross-Entropy and MSE) are smooth and convex with respect to the network outputs.
> This ensures that the auxiliary confidence loss provides a stable, well-behaved gradient signal during backpropagation.
>
>
> # Response to Weakness 3:
>
> To clarify, the pre-imputation stage is intentionally completely agnostic to the specific sample.
> Its sole objective is to establish a generic modality representation that resides safely within the correct topological space of the missing modality (respecting the "modality gap" phenomenon). Therefore, instance-specific semantics are not "destroyed" during pre-imputation; rather, they are intentionally deferred to post-imputation.
>
>
> We hypothesise that an imputed modality must consist of two distinct parts:
>
> 1. Modality-Specific Information (Global): Provided by the pre-imputation stage.
>
> 2. Instance-Specific Information (Local): Provided exclusively by the post-imputation stage.
>
> The instance-specific features and logical structure are recovered during the second-stage refinement (Sparse Cross-Attention).
> Here, the model actively retrieves instance-level cues from available modalities and injects them into the pre-imputed "canvas" as indicated in post-imputation. Our visualisation in Figure 4 also suggested that the imputed data fall in the distribution of its own modality.
>
>
>
> # Response to Weakness 4:
>
> We thank the reviewer for providing another insightful work to our attention. The suggested work was uploaded to arxiv on 9 Feb 2026 and accepted by CVPR 2026. This makes us unlikely to cite the paper before ICML submission ddl. We find the paper is very insightful, so we will cite it in the revised manuscript.
>
>
> # Response to Weakness 5:
>
> **It is learning an instance-level proxy for correctness, and this very process is what actively drives and enforces expert specialization as shown in our expert selection plots in Figure 6.**
>
> These two concepts are not mutually exclusive in our framework; rather, the former is the mechanism for the latter.
>
> In ConfSMoE, the router is indeed learning a proxy for correctness. specifically, it learns to estimate "how confident will Expert $i$ be on this specific token $x_i$?" Because different experts are initialised differently and reside in different sub-spaces, an expert will naturally produce higher confidence on subsets of data that align with its current weights (e.g., Expert A might naturally perform better on Text-heavy inputs, while Expert B performs better on Audio-heavy inputs, while Expert C may perform better on both modalities but semantic-heavy inputs).
>
> By using this confidence proxy for routing, ConfNet inherently directs specific types of data to the experts most capable of handling them.

---

> > ### Author Rebuttal · Reviewer_o3Di · 2026-04-04
> >
> > thanks for the rebuttal.
> >
> > as someone who once worked in the same domain, i strongly suggest authors read PID and its related work from paul liang and this would help strengthen the theoretical understanding of modality info. A combination with missingness can be an elegant grounding of ur work or inspire some good future works.
> >
> > Reviewer 5Krs also asked a good question and the authors handled it well.
> >
> > i'd consider all my concerns resolved and raise score to accept.

---

> > > ### Author Response · Authors · 2026-04-04
> > >
> > > Thanks for the constructive suggestion on modality understanding. We have noted the relevant papers. They are offering an unique perspective that inspires us to further exploration.
> > >
> > > All the best :D

---

### Decision · Program_Chairs · 2026-04-30

**Decision:**

Accept (regular)

**Comment:**

This paper explores gating mechanisms in sparse MoEs for handling multimodal input from arbitraries. The overall writing is concise and clear.

During the discussion, the paper primarily encountered some expression issues, and theoretically, there are still areas for improvement and revision.
+ In particular, reviewer o3Di raised questions about motivation, research principles, and validity, which the authors ultimately addressed during the rebuttal process.
+ Furthermore, reviewer P388 pointed out poor writing, numerous formatting and writing errors, and a lack of theoretical support.
+ Reviewer 5Krs also noted a disconnect between the author's main motivation, narration, and expression, as well as poor formatting.

The area chair agreed with the reviewers' comments, and the paper received a full positive score, bordering on acceptance. The authors also provided effective rebuttals, but the paper exhibited significant self-assessment discrepancies. The area chair fully considered this and identified numerous writing shortcomings, recommending rigorous revisions during the camera-ready phase.

Overall, the paper is on the verge of acceptance; the overall feedback has been good, but it still needs further improvements in the final version.